# Discrepancy-Based Active Learning for Domain Adaptation

**Antoine de Mathelin**[1,2]**, François Deheeger**[1]**, Mathilde Mougeot**[3,2]**, Nicolas Vayatis**[2] *

[1]Michelin, [2]Centre Borelli, Université Paris-Saclay, CNRS, ENS Paris-Saclay, [3]ENSIIE

## Abstract

The goal of the paper is to design active learning strategies which lead to domain adaptation under an assumption of Lipschitz functions. Building on previous work by Mansour et al. (2009) we adapt the concept of discrepancy distance between source and target distributions to restrict the maximization over the hypothesis class to a localized class of functions which are performing accurate labeling on the source domain. We derive generalization error bounds for such active learning strategies in terms of Rademacher average and localized discrepancy for general loss functions which satisfy a regularity condition. A practical K-medoids algorithm that can address the case of large data set is inferred from the theoretical bounds. Our numerical experiments show that the proposed algorithm is competitive against other state-of-the-art active learning techniques in the context of domain adaptation, in particular on large data sets of around one hundred thousand images.

## 1 Introduction

Machine learning models trained on a labeled data set from a *source* domain may fail to generalize on new *target* domains of interest (Saenko et al., 2010). This issue, which can be caused by *domain shift*, can be handled when no target labels are available through unsupervised domain adaptation methods (Ganin et al., 2016). Using a small sample of labeled target data can, besides, greatly improve the model performances (Motiian et al., 2017). Acquiring such new labels is often expensive (Settles, 2010) and one seeks to query as few labels as possible. This explains why strategies of optimal labels acquisition, referred as *active learning* (Cohn et al., 1994) seem very promising for domain adaptation (Su et al., 2020).

Active learning is a challenging task and a broad literature exists. On the one hand, some active learning methods introduce heuristic approaches which provide the benefit of using practical algorithm based on simple criteria. For instance, the spatial coverage of the target domain (Hu et al., 2010; Bodó et al., 2011) or the minimization of target uncertainties (RayChaudhuri & Hamey, 1995; Gal et al., 2017) are considered, as well as combination of these heuristics (Wei et al., 2015; Kaushal et al., 2019). However finding the proper heuristics is not straightforward and previous methods do not link their query strategy with the target risk (Viering et al., 2019). On the other hand, active learning methods based on distribution matching aim at minimizing a distribution distance between the labeled set and the unlabeled one (Balcan et al., 2009; Wang & Ye, 2015; Viering et al., 2019). These methods provide theoretical guarantees on the target risk through generalization bounds. However the computation of the distances is either not scalable to large-scale data sets (Balcan et al., 2009; Wang & Ye, 2015; Viering et al., 2019) or based on adversarial training (Su et al., 2020; Shui et al., 2020) which involves complex hyper-parameter calibration (Kurach et al., 2019).

In this work, we propose to address the issue of active learning for general loss functions under domain shift through a distribution matching approach based on discrepancy minimization (Mansour et al., 2009). In Section 2, we derive theoretical results by adopting a localized discrepancy distance (Zhang et al., 2020) between the labeled and unlabeled empirical distributions. This localized discrepancy is defined as the plain discrepancy considered on a hypothesis space restricted to hypotheses close to the labeling function on the labeled data set. This distance has the benefit to focus only on relevant

---

*Correspondence at `antoine.de-mathelin-de-papigny@michelin.com`

candidates for approximating the labeling function and thus provides tighter bound of the target risk under some given assumptions (Zhang et al., 2020). Based on this distance, we provide a generalization bound of the target risk involving pairwise distances between sample points (Theorem 1). Inspired by this generalization error bound, we propose in Section 3 an accelerated K-medoids query algorithm which scales to large data set. In Section 4, we present the related works and analytically show that our proposed approach displays tighter theoretical control of the target risk than the one provided by recent active learning methods. We finally present in Section 5 the benefit of the proposed approach on several empirical regression and classification active learning problems in the context of domain adaptation.

## 2 DISCREPANCY BASED ACTIVE LEARNING

Setup and theory develop in this section mainly focus on regression tasks. Section 4.3 presents how the algorithm derived from the theoretical results can be extended to classification tasks.

### 2.1 SETUP AND DEFINITIONS

Given two subsets $\mathcal{X} \subset \mathbb{R}^p$ and $\mathcal{Y} \subset \mathbb{R}^q$ and $d : \mathcal{X} \times \mathcal{X} \to \mathbb{R}_+$ a distance on $\mathcal{X}$, we denote the source data set $\mathcal{S} = \{x_1, ..., x_m\} \in \mathcal{X}^m$ and the target data set $\mathcal{T} = \{x'_1, ..., x'_n\} \in \mathcal{X}^n$. We consider the *domain shift* setting where the respective data sets $\mathcal{S}$ and $\mathcal{T}$ are drawn according to two different distributions $Q$ and $P$ on $\mathcal{X}$. We consider a loss function $L : \mathcal{Y} \times \mathcal{Y} \to \mathbb{R}_+$ and a hypothesis space $H$ of $k$-Lipschitz functions from $\mathcal{X}$ to $\mathcal{Y}$. We denote by $\mathcal{L}_D(h, h') = \mathrm{E}_{x \sim D}[L(h(x), h'(x))]$ the average loss (or risk) over any distribution $D$ on $\mathcal{X}$ between two hypotheses $h, h' \in H$. We also define the expected Rademacher complexity of $H$ for the distribution $P$ as:

$$\mathfrak{R}_n(H) = \mathop{\mathrm{E}}_{\{x'_i\}_i \sim P} \left[ \mathop{\mathrm{E}}_{\{\sigma_i\}_i \sim U} \left[ \sup_{h \in H} \frac{1}{n} \sum_{i=1}^{n} \sigma_i h(x'_i) \right] \right],$$

with $\sigma_i$ drawn according to $U$ the uniform distribution on $\{-1, 1\}$.

We consider a labeling function for each distribution: $f_Q : \mathcal{X} \to \mathcal{Y}$ and $f_P : \mathcal{X} \to \mathcal{Y}$. For adaptation to be possible, the two labeling functions are supposed to be close (Mansour et al., 2009). We finally consider the single-shot batch active learning framework (Viering et al., 2019) where all queried data are picked in one single batch of fixed budget of $K > 0$ queries. In this framework, an active learning algorithm takes as inputs the source data set $\mathcal{S}$ along with its corresponding recorded labels as well as the target data set $\mathcal{T}$. The algorithm then returns a batch of $K$ queried target data denoted $\mathcal{T}_K \subset \mathcal{T}$. The corresponding labels for $\mathcal{T}_K$ are then recorded through an oracle and used along with the source labeled data to fit an hypothesis $h \in H$. We denote by $\mathscr{L}_K = \mathcal{S} \cup \mathcal{T}_K$ the labeled data set. The goal is to select the $K$ target data to label in order to minimize the target risk of $h : \mathcal{L}_P(h, f_P)$.

### 2.2 LOCALIZED DISCREPANCY

To formulate the problem of active learning under domain shift as a distribution matching problem, one needs to consider a measure of divergence between distributions. Recent interest focuses on the discrepancy (Mansour et al., 2009) which proves to be useful for domain adaptation (Cortes & Mohri, 2014; Zhang et al., 2019) and is recently used on the active learning setting (Viering et al., 2019). However this metric, defined as a maximal difference between domain losses over the whole hypothesis space, is relatively conservative as it includes hypotheses that the learner might not ever consider as candidates for the labeling function (Cortes et al., 2019c; Zhang et al., 2020). Based on this consideration, we introduce a localized discrepancy (Zhang et al., 2020) to restrict the measure of divergence between domains on a set of relevant candidate hypotheses for approximating the labeling function:

**Definition 1.** *Localized Discrepancy.* Let $K > 0$ be the number of queries and $\epsilon \geq 0$. Let $\mathcal{T}_K$ be a queried batch of size $K$, the empirical distributions of $\mathscr{L}_K = \mathcal{S} \cup \mathcal{T}_K$ and $\mathcal{T}$ are respectively denoted $\widehat{Q}_K$ and $\widehat{P}$. Let $H$ be a hypothesis space and $L$ a loss function. The localized discrepancy is defined as:

$$disc_{H_\epsilon^K}(\widehat{Q}_K, \widehat{P}) = \max_{h, h' \in H_\epsilon^K} |\mathcal{L}_{\widehat{Q}_K}(h, h') - \mathcal{L}_{\widehat{P}}(h, h')|, \tag{1}$$

*with* $H_\epsilon^K = \{h \in H; L(h(x), f_Q(x)) \leq \epsilon \ \forall x \in \mathscr{L}_K\}$

The localized space $H_\epsilon^K$ includes hypotheses "consistent" with $f_Q$ on the labeled data, i.e. hypotheses fitting the labeled data with an error below $\epsilon$. The parameter $\epsilon$ drives the size of $H_\epsilon^K$. Obviously, $\epsilon$ needs to be high enough to ensure that $H_\epsilon^K$ is not an empty subset. Under appropriate assumptions on $\epsilon$, a preliminary result is an empirical target risk bound for the localized discrepancy:

**Proposition 1.** *Let $K > 0$ be the number of queries and $H$ a hypothesis space. Let $\widehat{P}$ and $\widehat{Q}_K$ be the empirical distributions of the respective sets $\mathcal{T}$ and $\mathscr{L}_K = \mathcal{S} \cup \mathcal{T}_K$ of respective size $n$ and $m + K$. We assume that $L$ is a symmetric, $\mu$-Lipschitz and bounded loss function verifying the triangular inequality. We denote by $M$ the bound of $L$. Let $\eta_H$ be the ideal maximal error on $\mathcal{S} \cup \mathcal{T}$:*

$$\eta_H \triangleq \min_{h \in H} \max_{x \in \mathcal{S} \cup \mathcal{T}} [L(h(x), f_Q(x)) + L(h(x), f_P(x))] \tag{2}$$

*Then, for any $\epsilon \geq \eta_H$, any hypothesis $h \in H_\epsilon^K$ and any $\delta > 0$, the following generalization bound holds with at least probability 1-$\delta$:*

$$\mathcal{L}_P(h, f_P) \leq \mathcal{L}_{\widehat{Q}_K}(h, f_Q) + \mathrm{disc}_{H_\epsilon^K}(\widehat{Q}_K, \widehat{P}) + \eta_H + 2\mu\mathfrak{R}_n(H) + M\left(\sqrt{\frac{\log(\frac{1}{\delta})}{2n}}\right). \tag{3}$$

This bound share a similar form than the one derived in Cortes et al. (2019c) for the discrepancy. The ideal maximal error $\eta_H$ characterizes the difficulty of the adaptation problem as well as the ability to learn the labeling functions with the considered hypothesis set $H$. If $f_P$ and $f_Q$ significantly differ on $\mathcal{S} \cup \mathcal{T}$, $\eta_H$ will be high and the proposed bound will lack of informativeness. We then follow the common domain adaptation assumption that the difference between the two functions is small (Mansour et al., 2009; Zhang et al., 2020).

## 2.3 MAIN RESULTS: GENERALIZATION BOUNDS FOR ACTIVE LEARNING

Considering the previous bound (Proposition 1) it appears that a natural way of choosing the $K$ queries in an active learning perspective is to pick the target data minimizing the localized discrepancy. Unfortunately this is a difficult problem for an arbitrary functional space $H$, since it leads to compute a maximum over the set space $H_\epsilon^K$. Our main idea is then to further bound the localized discrepancy with a computable criterion:

**Theorem 1.** *Let $K > 0$ be the number of queries, $H$ a hypothesis space of $k$-Lipschitz functions and $\epsilon \geq \eta_H$. Let $\mathscr{L}_K = \mathcal{S} \cup \mathcal{T}_K$ be the labeled set and $\mathcal{T}$ the target set drawn according to $P$. We assume that $L$ is a symmetric, $\mu$-Lipschitz and bounded loss function verifying the triangular inequality. We define $M$ such that $L(y, y') \leq M$ for any $y, y' \in \mathcal{Y}$. For any hypothesis $h \in H_\epsilon^K$ and any $\delta > 0$, the following generalization bound holds with at least probability 1-$\delta$:*

$$\mathcal{L}_P(h, f_P) \leq \mathcal{L}_{\widehat{Q}_K}(h, f_Q) + \frac{2k\mu}{n} \sum_{x' \in \mathcal{T}} d(x', \mathscr{L}_K) + 2\epsilon + \eta_H + 2\mu\mathfrak{R}_n(H) + \sqrt{\frac{M^2 \log(\frac{1}{\delta})}{2n}} \tag{4}$$

*With $d(x', \mathscr{L}_K) = \min_{x \in \mathscr{L}_K} d(x', x)$ and $d(x', x)$ the distance from $x$ to $x'$.*

Visual insights to understand Theorem 1 are presented in Figure 1 : the main idea is to approximate the maximal hypotheses $h, h' \in H_\epsilon^K$ returning the localized discrepancy by the $k$-Lipschitz envelope of the labeling function $f_Q$, consistent with $f_Q$ on the labeled points, i.e. at most $\epsilon$ close to $f_Q$ on these points. Indeed, for any target point $x' \in \mathcal{T}$, the gap between $h, h'$ on $x'$ : $L(h'(x'), h(x'))$ is upper bounded by twice the distance from $x'$ to its closest labeled point times the Lipschitz constants of $L, h$ and $h'$ plus the error on the labeled point : $L(h'(x'), h(x')) \leq 2k\mu\, d(x', \mathscr{L}_K) + 2\epsilon$.

The generalization bound of Theorem 1 highlights the trade-off that exists between the Lipschitz constant $k$ of the hypothesis space $H$ and the parameters $\eta_H$ and $\epsilon$. To reduce the term $2\epsilon + \eta_H$ in order to obtain tighter controls over the target risk, one needs to consider a more complex set of hypothesis and thus to increase the Lipschitz constant $k$. Then, to benefit from the theoretical guarantees of this bound, a careful choice of hypothesis set have to be made (cf Section 4.3).

The important benefit of the derived bound is to bring out the bounding criterion $\sum_{x' \in \mathcal{T}} d(x', \mathscr{L}_K)$ which is independent of the hypothesis complexity (characterized by $k$) and the loss function, it only involves pairwise distances between sample points. As this criterion is computable and depends on the queried batch $\mathcal{T}_K$, we can then propose an active learning strategy.

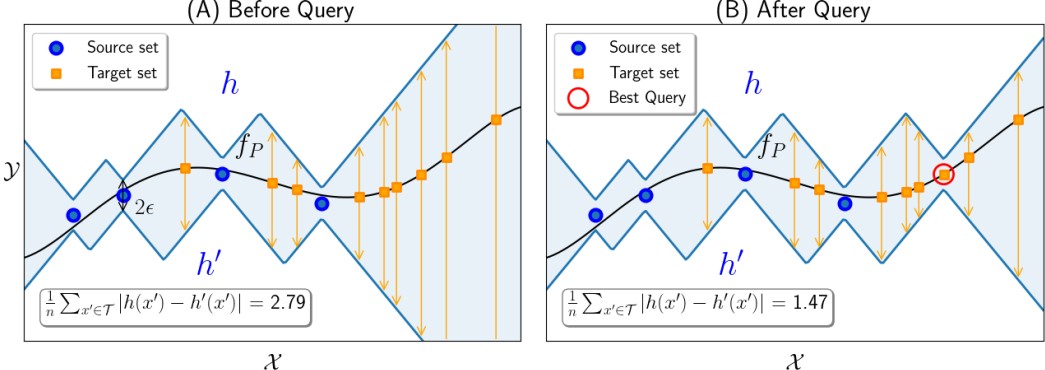

Figure 1: **Visual insights of Theorem 1** : Any potential candidate for $f_P$ in $H_\epsilon^K$ returns values between $h$ and $h'$ : two hypotheses $\epsilon$ close to $f_Q$ on the labeled points and with slopes of factor $k$ everywhere. Thus, an approximation of the localized discrepancy is given by the mean of gaps between $h$ and $h'$ (mean length of orange arrows) which can be approximated by the distance to the labeled set in $\mathcal{X}$ times $k$. The best target point to label is chosen in order to minimize this sum.

## 3   DISCREPANCY-BASED ALGORITHM

Theorem 1 directly implies that selecting the $K$ queries minimizing $\sum_{x' \in \mathcal{T}} d(x', \mathcal{L}_K)$ leads to minimize an upper bound of the target risk. We can thus propose an algorithm dedicated to active learning which provides theoretical guarantees on the target risk.

Seeking the $K$ queries minimizing $\sum_{x' \in \mathcal{T}} d(x', \mathcal{L}_K)$ corresponds to solve a K-medoids problem (Kaufmann & Rousseeuw, 1987). Notice however that it does not consist of a K-medoids performed directly on the target domain as the source data are already labeled and considered as medoids. The presented algorithm only differs in the initialization process, where the distance between each target and its nearest source neighbour needs to be computed.

Several algorithms exist to solve or approximate the K-medoids (Kaufman & Rousseeuw, 2009), (Ng & Han, 2002), (Park & Jun, 2009). Here we use the greedy version of the algorithm. It can be shown that the gain of the greedy algorithm (the amount of decrease of the criterion by selecting $K$ points) is at least a $(1 - 1/e)$-approximation of the optimal gain (cf Appendix).

It is well known that K-medoids algorithms suffer from computational burdens or memory issues on large and moderately large data sets ($\sim$ 100K data) (Newling & Fleuret, 2017). Indeed they require to compute huge pairwise distance matrix between the source and target samples as well as between targets. Precisely, the greedy K-medoids algorithm presents a complexity of $\mathcal{O}(p(nm + n^2) + Kn^2)$ and a memory usage of $\mathcal{O}(nm + n^2)$ with $m, n, K$ the respective size of $\mathcal{S}, \mathcal{T}$ and $\mathcal{T}_K$. $p$ is the dimension of $\mathcal{X}$.

To handle this issue, we propose an adaptation of the K-medoids greedy algorithm with better scalability (Algorithm 1). This algorithm performs the following steps:

1) Computation of the distance to the closest source for each target points via a KD-trees random forest algorithm (Silpa-Anan & Hartley, 2008) of complexity $\mathcal{O}(T(m + pn)\log(m)^2)$ with $T$ the number of trees.

2) Medoids initialization using the greedy algorithm on a random target batch of $B$ samples with complexity $\mathcal{O}((K + p)B^2)$.

3) An iterative algorithm on the model of (Park & Jun, 2009), combining assignment of each target point to its closest medoid ($\mathcal{O}(Kpn)$) and medoid update inside each cluster.

4) The medoid update for each cluster is done through an original Branch-and-Bound algorithm (Land & Doig, 2010) which estimates the criterion by iteration over mini-batch, target points for which criterion is bigger than a statistical threshold are left aside. Thus, the number of pairwise

distances to compute is reduced at each iteration until the maximal iteration number is reached or all target points are left aside. Under some assumptions, we can show that the complexity of the update for all cluster is $\mathcal{O}(n^{3/2}K^{-1/2})$ (the proof is given in the supplementary material).

The overall complexity is then $\mathcal{O}(T(m+pn)\log(m)^2 + (K+p)B^2 + Kpn + pn^{3/2}K^{-1/2})$ which provides reasonable computational time for moderately large data set ($n, m \sim 10^5$ and $p \sim 10^3$). Empirical comparison of computational time is also provided in the supplementary material.

---

**Algorithm 1** Accelerated K-medoids

1: $\{d(x', \mathcal{S})\}_{x' \in \mathcal{T}} \leftarrow$ **KDT-Forest-Nearest-Neighbour**$(\mathcal{S}, \mathcal{T})$   # *Compute distances to source*
2: $\mathcal{T}_K = \{x_i^*\}_{i=1..K} \leftarrow$ **Kmedois-Greedy**$(\mathcal{T}, B, K, \{d(x', \mathcal{S})\}_{x' \in \mathcal{T}})$   # *Initialize medoids*
3: $C_i \leftarrow \{x' \in \mathcal{T} \,|\, d(x', x_i^*) = \min\limits_{j=1..K} d(x', x_j^*) \wedge d(x', \mathcal{S})\}$   # *Assign targets to closest cluster*
4: **while** *any medoid $x_i^*$ can be updated* **do**
5:    **for** $i$ from 1 to $K$ **do**
6:       $x_i^* = \underset{x' \in C_i}{\operatorname{argmin}} \sum\limits_{x'' \in C_i} d(x', x'') \leftarrow$ **Branch-and-Bound**$(C_i)$   # *Update medoid*
7:    **end for**
8:    For all $x' \in \mathcal{T}$, compute $d(x', x_i^*)$ for all updated medoids and reassign $x'$ to the closest $C_i$
9: **end while**

---

# 4 RELATED WORK AND DISCUSSION

## 4.1 RELATED WORK

*Active learning as distribution matching.* Active learning methods based on distribution matching aim at reducing the gap between the distributions of the labeled sample and the unlabeled one with a minimal query budget. Several metrics are used to measure the gap between distributions as the Transductive Rademacher Complexity (Gu & Han, 2012), the MMD (Wang & Ye, 2015; Kim et al., 2016; Viering et al., 2019) the Disagreement Coefficient (Hanneke, 2007; Balcan et al., 2009; Beygelzimer et al., 2009; Cortes et al., 2019a;b; 2020), the $\mathcal{H}$-divergence (Sinha et al., 2019; Gissin & Shalev-Shwartz, 2019; Su et al., 2020) or the Wasserstein distance (Shui et al., 2020). To the best of our knowledge, only one paper deals with the discrepancy for active learning (Viering et al., 2019). The authors consider the discrepancy on the space of RKHS hypotheses with PSD kernels and provide an explicit way of computing the discrepancy using eigen-value analysis. However, the corresponding algorithm encounters computational burden and could hardly be applied on large sets.

*K-medoids for active learning.* Many active learning methods use a K-medoids algorithm as an heuristic measure of representativeness (Lin et al., 2009; Gomes & Krause, 2010; Wei et al., 2013; Iyer & Bilmes, 2013; Zheng et al., 2014). The K-medoids is in general computed on a smaller set of selected targets with the higher uncertainties (Wei et al., 2015; Kaushal et al., 2019). In this present work, we provide theoretical insights for this algorithm by highlighting the link with discrepancy minimization.

*Active learning for domain adaptation.* Our work is related to the recent advances on active learning for domain adaptation as we also consider the domain shift hypothesis (Rai et al., 2010; Saha et al., 2011; Deng et al., 2018; Su et al., 2020). These works use in general the output of a domain classifier to measure the informativeness of target samples. In our work, we consider instead the distance to the source sample to capture informative target data.

*Lipschitz consistent functions for active learning.* In the context of function optimization, some methods consider the set of Lipschitz or locally Lipschitz functions consistent with the observations (Valko et al., 2013; Grill et al., 2015; Malherbe & Vayatis, 2017). We use similar functions to approximate the maximal hypotheses returning the localized discrepancy. Notice that the goal of the previous papers differ from ours as they aim at finding the maximum of the labeling function.

## 4.2 COMPARISON WITH EXISTING GENERALIZATION BOUNDS FOR ACTIVE LEARNING

Some previous works on active learning also propose theoretical bound on the target risk. In this section, we will compare them with our derived bound of equation (4) in order to relate our

contribution with existing works. As their framework is the pure active learning setting which differs from the setting consider here, we make the comparisons under the simplifying assumptions: $f = f_P = f_Q$, $f \in H$ and $\epsilon = 0$. In this case, our bound is written:

$$\mathcal{L}_P(h, f) \leq (2k\mu/n) \sum_{x' \in \mathcal{T}} d(x', \mathscr{L}_K) + 2\mu\mathfrak{R}_n(H) + \sqrt{M^2 \log(1/\delta)/2n} \tag{5}$$

The paper by Sener & Savarese (2018) proposes the K-centers algorithm for active learning based on an easily computable criterion offering theoretical guarantees. For a regression loss $L$ and under the previous assumptions, it controls the target risk as follows:

$$\mathcal{L}_P(h, f) \leq 2k\mu \max_{x' \in \mathcal{T}} d(x', \mathscr{L}_K) + 2\mu\mathfrak{R}_n(H) + \mathcal{O}\left(\sqrt{M^2 \log(1/\delta)/2n}\right) \tag{6}$$

We directly observe that this bound is looser than the one of equation (5). Indeed, in our case the target risk is controlled with the mean of distances between unlabeled points and the labeled set whereas K-centers considers the maximum of these distances. Notice however that both algorithms use greedy selection approximation. Thus, in some cases, the queries from K-centers may lead to a smaller bound than the ones from K-medoids.

A generalization bound for active learning involving the Wasserstein distance $W_1$ has also been proposed (Shui et al., 2020). An adaptation of this bound for the proposed scenario with the aforementioned assumptions could be written as follows:

$$\mathcal{L}_P(h, f) \leq 2k\mu W_1(\widehat{Q}_K, \widehat{P}) + 2\mu\mathfrak{R}_n(H) + \mathcal{O}\left(\sqrt{M^2 \log(1/\delta)/2n}\right)$$

Where $W_1(\widehat{Q}_K, \widehat{P}) = \arg\min_{\gamma \in \Gamma} \sum_{x \sim \widehat{Q}_K} \sum_{x' \sim \widehat{P}} \gamma_{xx'} d(x, x')$ with $\Gamma = \{\gamma \in \mathbb{R}^{n \times (m+K)} ; \gamma\mathbf{1} = \frac{1}{n}\mathbf{1} ; \gamma^T\mathbf{1} = \frac{1}{m+K}\mathbf{1}\}$.

Thus, in observing that $d(x, x') \geq d(x', \mathscr{L}_K)$ for any $x' \in \mathcal{T}$ and $x \in \mathscr{L}_K$ we can show that our presented bound of equation (5) is tighter than the one above.

## 4.3 DISCUSSION ABOUT THE ASSUMPTIONS AND LIMITATIONS

One crucial point of the present work is the setting of the $\epsilon$ parameter of the hypothesis set $H_\epsilon^K$. In practice, the parameter $\epsilon$ is determined by the hypothesis set and the training algorithm that the learner considers. For instance, if the learner uses over-parameterized hypotheses overfitted on the labeled data set, the parameter $\epsilon$ will be small because for any $h$, $h(x) \simeq f_Q(x)$ on $\mathscr{L}_K$. This could leads to $\epsilon < \eta_H$ and the bound would not be valid anymore. This highlights the trade-off between fitting the source data and generalizing to the target domain. As, in practice, the parameter $\eta_H$ is hard to estimate, choosing larger $\epsilon$ (by regularizing the hypotheses) is safer, but will lead to larger bounds.

To deal with this difficulty, we use, in our experiments, a set of neural networks $H$ regularized through weight clipping. The value of the clipping parameter is directly linked to the Lipschitz constant of $H$. By selecting an adequate clipping parameter and network architecture, we ensure that $H$ is complex enough to learn the task on the source domain but sufficiently regularized to avoid over-fitting (and thus avoid $\epsilon < \eta_H$). For this purpose, we select the architecture and the clipping parameter through validation on the source labeled data.

Regularity assumptions on the loss function $L$ are essentially verified by norms as the $L_p$ which are common losses for regression problems. However, they are are not verified by most classification losses. In fact, classification loss as the cross-entropy is bounded between 0 and 1 and can not increase linearly with the distance to the closest labeled point. In this context, considering target points far away from sources as informative points is not efficient. In fact, the most interesting points are the ones in the margin between classes (Balcan et al., 2007). Thus, in order to focus the K-medoids selection in the margin, and thus extent the proposed algorithm to classification task, we propose an improved version of our algorithm, the **Weighted K-medoids (K-medoids+W)**. This algorithm performs the K-medoids algorithm with a weighted criterion. To compute the weights, we consider the Best-vs-Second-Best (BVSB) criterion (Joshi et al., 2009) which is the difference between the probabilities of the best class and the second best class, given by a hypothesis pre-trained on the source data set. The Weighted K-medoids optimization can be written as follows:

$$\min_{\mathcal{T}_K} \sum_{x' \in \mathcal{T}} \text{bvsb}(x') \min_{x \in \mathcal{S} \cup \mathcal{T}_K} d(x, x') \tag{7}$$

## 5 EXPERIMENTS

We choose to compare the performances of our algorithm to classical active learning methods on regression and classification problems in a domain shift context. We consider the single-batch active learning setting (Viering et al., 2019) where all queries are taken at the same time in one batch. We compare the results obtained on the target domain for different query and training methods. The experiments have been run on a (2.7GHz, 16G RAM) computer. The source code is provided on GitHub [1]. We use the open source code of the corresponding authors for BADGE (Ash et al., 2019) and the implementations from ADAPT[2] (de Mathelin et al., 2021) for the domain adaptation methods.

### 5.1 COMPETITORS

We compare the proposed approach with the following query methods : **Random Sampling**; **K-means** (Hu et al., 2010); **K-centers** (Sener & Savarese, 2018); **Diversity** (Jain & Grauman, 2016); **QBC** (RayChaudhuri & Hamey, 1995); **BVSB** (Joshi et al., 2009); **AADA** (Su et al., 2020) : an hybrid active learning method for domain adaptation using a combination of entropy measure from a classifier and the outputs of a domain discriminator; **BADGE** (Ash et al., 2019): an hybrid deep active learning method optimizing for both uncertainty and diversity. **CLUE** (Prabhu et al., 2020) an active domain adaptation strategy that select instances that are both uncertain and diverse.

We select four different training methods **Uniform Weighting**; **Balanced Weighting** : Assign balanced total weight between source and target instances; **TrAdaBoost** (Pardoe & Stone, 2010) : Transfer learning regression method based on a reverse boosting principle; **Adversarial training** : unsupervised features transformation on the model of DANN (Ganin et al., 2016).

To make a fair comparison between the different query strategies, we use for all experiments, the same set of training hypotheses $H$. We define $H$ as the set of neural networks composed of two fully connected hidden layers of 100 neurons, with ReLU activations and projection constraints on the layer norms ($< 1.$). We use the Adam optimizer (Kingma & Ba, 2015). The network architecture is defined to be complex enough that the network provides a good approximation of the labeling function on the source domain in all experiments. Besides, for each experiment, fine-tuning of the optimization hyper-parameters (epochs, batch sizes...) is performed using only source labeled data. We assume that the architecture and the resulting hyper-parameters will still be appropriate after adding the queried target data to the training set (see Section 4.3). Finally, for distance-based algorithm, we consider the $L_1$ distance computed in the penultimate layer of a network pre-trained on sources. We use an ensemble of 10 models in QBC and the greedy version of K-centers.

### 5.2 SUPERCONDUCTIVITY DATA SET

As there is very few public data sets for domain adaptation with regression tasks (Teshima et al., 2020), we choose an UCI data set with a reasonable amount of instances and split it in different domains using the setup of (Pardoe & Stone, 2010). We choose *Superconductivity* (Hamidieh, 2018; Dua & Graff, 2017) which is composed of features extracted from superconductors chemical formula. The task consists in predicting their critical temperature.

*Experimental setup:* The data set is divided in four domains following (Pardoe & Stone, 2010) : low (*l*), middle-low (*ml*), middle-high (*mh*) and high (*h*) of around 4000 instances and 166 features. We use a learning rate of 0.001, a number of epochs of 100, a batch size of 128 and the mean squared error as loss function. We conduct an experiment for the 12 pairs of domains. We vary $K$ from 5 to 300 and repeated each experiment 8 times with different random seeds. We report the mean absolute error (MAE) on the target unlabeled data for all experiments when $K = 20$ in Table 1. We present the MAE evolution for the adaptation from *mh* to *h* in Figure 2.

*Results:* We observe on Figure 2 that, for any $K > 0$, the K-medoids algorithm presents the lowest MAE on the target data for the three different training methods. In particular we observe a significant performance gain of K-medoids when using TrAdaBoost which provides the lowest MAE for the majority of fixed budget $K$ compared to other training methods. These observations are confirmed on Table 1 where we observe that K-medoids presents the lowest MAE in 11 experiments over 12. We

---

[1]`https://github.com/antoinedemathelin/dbal`
[2]`https://github.com/adapt-python/adapt`

Table 1: MAE on the critical temperature for the Superconductivity experiments with Balanced Weighting and $K = 20$. Standard deviation are given in the supplementary material.

| Experiment | l→ml | l→mh | l→h | ml→l | ml→mh | ml→h | mh→l | mh→ml | mh→h | h→l | h→ml | h→mh |
|---|---|---|---|---|---|---|---|---|---|---|---|---|
| Random | 15.33 | 15.80 | 17.45 | 16.53 | 11.39 | 14.70 | 17.65 | 12.83 | 10.36 | 18.75 | 14.86 | 10.54 |
| Kmeans | 14.43 | 13.60 | **13.98** | 15.79 | 10.19 | 12.73 | 17.18 | 12.67 | 10.02 | 22.10 | 14.69 | 9.76 |
| QBC | 20.00 | 19.03 | 20.08 | 15.89 | 12.24 | 15.31 | 20.78 | 12.87 | 10.19 | 31.88 | 18.86 | 10.65 |
| Kcenters | 19.21 | 15.73 | 16.85 | 15.75 | 11.62 | 13.44 | 22.17 | 12.74 | 10.24 | 36.50 | 19.60 | 10.39 |
| Diversity | 19.46 | 18.21 | 18.68 | 16.01 | 11.94 | 15.36 | 23.92 | 14.31 | 10.70 | 37.97 | 20.89 | 10.78 |
| Kmedoids | **12.70** | **13.57** | 14.11 | **14.49** | **10.02** | **12.52** | **15.36** | **12.37** | **9.79** | **16.62** | **14.14** | **9.32** |

also observe here that methods based on spatial consideration as K-medoids, K-means and K-centers select more informative target points than the uncertainty based method QBC. This comes from the fact that, in batch mode, QBC is selecting close target points with similar uncertainty level. Finally, K-medoids outperforms K-means because it takes into account the distance to source points and then queries less redundant information.

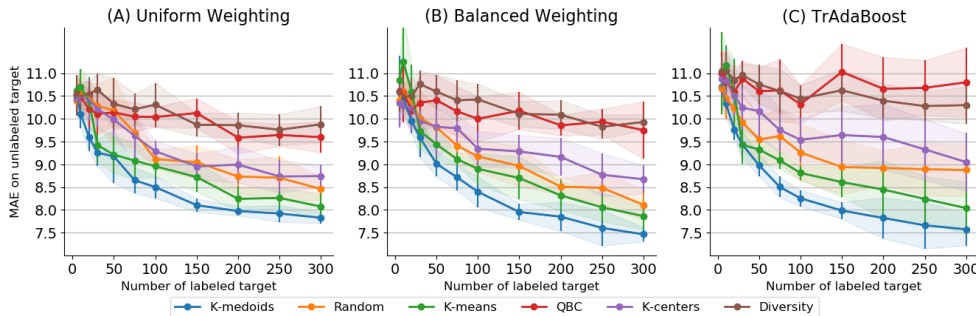

Figure 2: Results for the Superconductivity data set ($mh \rightarrow h$ experiment). Evolution of the MAE in function of the budget $K$ for three different training methods and six query methods.

## 5.3 OFFICE DATA SET

The office data set (Saenko et al., 2010) consists in pictures of office items coming from different domains: amazon or webcam. The task is a multi-classification problem with 31 classes (chairs, printers, ...). The goal is to use data from the amazon domain where labels are easily available to learn a good model on the webcam domain where a few labels are chosen using active learning methods.

*Experimental setup:* We consider the adaptation from "amazon" with 2817 labeled images to "webcam" with 795 unlabeled images. We use, as input features, the outputs of the ResNet-50 network (He et al., 2016) pretrained on ImageNet. We vary $K$ from 5 to 300, repeating each experiment 8 times. The learning rate is 0.001, the number of epochs 60 and the batch size 128.

*Results:* Figure 3.A presents the results obtained. We observe that the K-medoids+W algorithm provides the best performances for almost any $K$ and in particular for small values of $K$. We then present the visualization of the two first components of the PCA transform on Figure 4. We observe that the K-medoids+W algorithm queries points at the center of the target distribution but at a reasonable distance from the sources. The Random and K-means algorithms select a representative subset of the target distribution but without taking into account the sources and therefore query redundant information. K-centers selects data far from the source domain but which are less representative of the the target distribution.

## 5.4 DIGITS DATA SET

We consider the experiment proposed in (Ganin et al., 2016) where a synthetic digits data set: SYNTH is used to learn a classification task for a data set of real digits pictures: SVHN (Street-View House Number) (Netzer et al., 2011). Both data sets are composed of around 65k images of size $28 \times 28$.

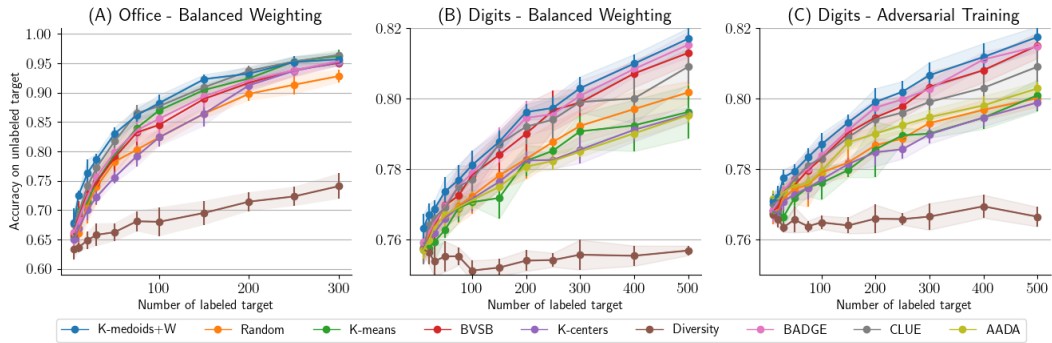

Figure 3: Office and digits results. Evolution of the accuracy in function of the budget $K$.

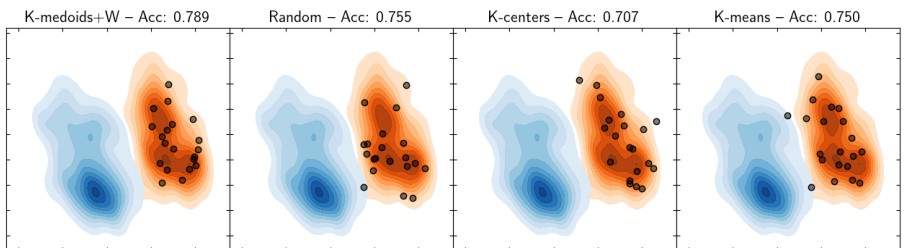

Figure 4: Visualization of the two PCA first components of the Office data set input space. Queries are reported with black points for each query method with $K = 20$.

*Experimental setup:* To handle the large number of data, we use the accelerated K-medoids algorithm (Algorithm 1) with $T = 50$ trees and a initial batch size of $B = 5000$. We use the KD-trees random forest nearest neighbour algorithm in Diversity and K-centers to approximate the distance of each target data to the source data set. We consider two kinds of input features: the ones obtained with the convulational part of a Lenet (LeCun et al., 1998) trained with the source labeled data and the ones coming from the same network but trained with adversarial training following the model of DANN (Ganin et al., 2016). In both cases, the network is pre-trained on 30 epochs with a batch size of 128 and a learning rate of 0.001, for the adversarial training the trade-off parameter $\lambda$ is set to 0.1 following the setup of (Su et al., 2020). After the query process, a Balance Weighting training is performed with the source and target labeled data using the same optimization hyper-parameters than before. Experiments are conducted 8 times for $K$ between 10 and 500.

*Results:* Figure 3.B and 3.C correspond to the evolution of accuracy with respect to $K$ for the experiments conducted with the features obtained respectively without and with adversarial training. We observe that, for any $K > 0$, K-medoids+W provides improved results over other query strategies in both cases. This highlights the ability of the method to select informative target points in a variety of scenarios.

## 6 CONCLUSION AND FUTURE WORK

This work introduces a novel active learning approach based on a localized discrepancy between the labeled and unlabeled distributions. We provide both theoretical guarantees of this approach and an active learning algorithm scaling to large data sets. Several experiments show very competitive results of the proposed approach. Future work will focus on considering a more appropriate distance on the input space, giving more importance to relevant features with respect to the task.

## REPRODUCIBILITY STATEMENT

To help the reproducibility of the results presented in this work, the source code of the experiments is available at `https://github.com/antoinedemathelin/dbal`.

Furthermore, the presented methods and the majority of the competitors have been implemented with "pythonic" objects which implement a *fit* and *predict* methods. Thus, the code can easily be used on other data sets than the ones considered in this work.

Finally, notebooks are provided in the repository to enable a rapid access to the methods and the possibility to try for different hyper-parameters.

## ACKNOWLEDGMENTS

Part of this research was funded by the Manufacture Française des Pneumatiques Michelin and the Industrial Data Analytics and Machine Learning chair of Centre Borelli from ENS Paris Saclay.

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

# Appendix

We recall here the notations and definitions used in the following:

- $\mathcal{X} \subset \mathbb{R}^p$ and $\mathcal{Y} \subset \mathbb{R}$ are the respective input and output subsets.
- $d : \mathcal{X} \times \mathcal{X} \to \mathbb{R}_+$ is a distance on $\mathcal{X}$.
- $P$ and $Q$ are two distributions on $\mathcal{X}$.
- $\mathcal{T} = \{x'_1, ..., x'_n\} \in \mathcal{X}^n$ is the unlabeled target data set and $\mathcal{S} = \{x_1, ..., x_m\} \in \mathcal{X}^m$ the labeled source data set drawn respectively from $P$ and $Q$.
- $L : \mathcal{Y} \times \mathcal{Y} \to \mathbb{R}_+$ is a loss function.
- $H$ is a hypothesis set of $k$-Lipschitz functions from $\mathcal{X}$ to $\mathcal{Y}$.
- $\mathrm{E}_{x \sim D}[L(h(x), h'(x))]$ is the average loss (or risk) over any distribution $D$ on $\mathcal{X}$ between two hypotheses $h, h' \in H$.
- $\mathfrak{R}_n(H) = \underset{\{x'_i\}_i \sim P}{\mathrm{E}} \left[ \underset{\{\sigma_i\}_i \sim U}{\mathrm{E}} \left[ \underset{h \in H}{\sup} \frac{1}{n} \sum_{i=1}^{n} \sigma_i h(x'_i) \right] \right]$, is the expected Rademacher complexity of $H$, with $U$ the uniform distribution on $\{-1, 1\}$.
- $f_Q : \mathcal{X} \to \mathcal{Y}$ is the source labeling function.
- $f_P : \mathcal{X} \to \mathcal{Y}$ is the target labeling function.
- $K > 0$ is the number of queries
- $\mathcal{T}_K \subset \mathcal{T}$ with $|\mathcal{T}_K| = K$ is a queried batch or subset.
- $\mathscr{L}_K = \mathcal{S} \cup \mathcal{T}_K$ is the labeled data set.
- $\widehat{Q}, \widehat{Q}_K$ and $\widehat{P}$ are the respective empirical distributions on $\mathcal{X}$ of $\mathcal{S}$, $\mathcal{S} \cup \mathcal{T}_K$ and $\mathcal{T}$.
- $H_\epsilon^K = \{h \in H; L(h(x), f_Q(x)) \leq \epsilon \ \forall x \in \mathscr{L}_K\}$ is the localized hypothesis space .
- $\mathrm{disc}_{H_\epsilon^K}(\widehat{Q}_K, \widehat{P}) = \max_{h, h' \in H_\epsilon^K} |\mathcal{L}_{\widehat{Q}_K}(h, h') - \mathcal{L}_{\widehat{P}}(h, h')|$ is the localized discrepancy between $\widehat{Q}_K$ and $\widehat{P}$.

## A   PROOF OF PROPOSITION 1

**Proposition 1.** Let $K > 0$ be the number of queries and $H$ a hypothesis space. Let $\widehat{P}$ and $\widehat{Q}_K$ be the empirical distributions of the respective sets $\mathcal{T}$ and $\mathscr{L}_K = \mathcal{S} \cup \mathcal{T}_K$ of respective size $n$ and $m + K$. We assume that $L$ is a symmetric, $\mu$-Lipschitz and bounded loss function verifying the triangular inequality. We denote by $M$ the bound of $L$. Let $\eta_H$ be the ideal maximal error on $\mathcal{S} \cup \mathcal{T}$:

$$\eta_H \triangleq \min_{h \in H} \max_{x \in \mathcal{S} \cup \mathcal{T}} [L(h(x), f_Q(x)) + L(h(x), f_P(x))] \tag{8}$$

Then, for any $\epsilon \geq \eta_H$, any hypothesis $h \in H_\epsilon^K$ and any $\delta > 0$, the following generalization bound holds with at least probability 1-$\delta$:

$$\mathcal{L}_P(h, f_P) \leq \mathcal{L}_{\widehat{Q}_K}(h, f_Q) + \mathrm{disc}_{H_\epsilon^K}(\widehat{Q}_K, \widehat{P}) + \eta_H + 2\mu \mathfrak{R}_n(H) + M \left( \sqrt{\frac{\log(\frac{1}{\delta})}{2n}} \right) . \tag{9}$$

*Proof.* Let's consider $h \in H$. According to (Mohri et al., 2018) we have for any $\delta > 0$, with probability at least $1 - \delta$:

$$\mathcal{L}_P(h, f_P) \leq \mathcal{L}_{\widehat{P}}(h, f_P) + 2\mu \mathfrak{R}_n(H) + M \sqrt{\frac{\log(\frac{1}{\delta})}{2n}} . \tag{10}$$

Besides, we have for any $h, h_0 \in H_\epsilon^K$:

$$
\begin{aligned}
\mathcal{L}_{\widehat{P}}(h, f_P) &= \mathcal{L}_{\widehat{Q}_K}(h, f_Q) + \mathcal{L}_{\widehat{P}}(h, f_P) - \mathcal{L}_{\widehat{Q}_K}(h, f_Q) \\
&\leq \mathcal{L}_{\widehat{Q}_K}(h, f_Q) + \mathcal{L}_{\widehat{P}}(h, h_0) + \mathcal{L}_{\widehat{P}}(h_0, f_P) - \mathcal{L}_{\widehat{Q}_K}(h, h_0) + \mathcal{L}_{\widehat{Q}_K}(h_0, f_Q) \\
&\leq \mathcal{L}_{\widehat{Q}_K}(h, f_Q) + \max_{h,h' \in H_\epsilon^K} |\mathcal{L}_{\widehat{P}}(h, h') - \mathcal{L}_{\widehat{Q}_K}(h, h')| + \mathcal{L}_{\widehat{P}}(h_0, f_P) + \mathcal{L}_{\widehat{Q}_K}(h_0, f_Q)
\end{aligned}
\tag{11}
$$

As the inequality is true for any $h_0 \in H_\epsilon^K$, we have in particular:

$$
\mathcal{L}_{\widehat{P}}(h, f_P) \leq \mathcal{L}_{\widehat{Q}_K}(h, f_Q) + \max_{h,h' \in H_\epsilon^K} |\mathcal{L}_{\widehat{P}}(h, h') - \mathcal{L}_{\widehat{Q}_K}(h, h')| + \min_{h_0 \in H_\epsilon^K} \left( \mathcal{L}_{\widehat{P}}(h_0, f_P) + \mathcal{L}_{\widehat{Q}_K}(h_0, f_Q) \right)
\tag{12}
$$

We notice that, for any $h_0 \in H_\epsilon^K$:

$$
\mathcal{L}_{\widehat{P}}(h_0, f_P) + \mathcal{L}_{\widehat{Q}_K}(h_0, f_Q) \leq \max_{x \in \mathcal{S} \cup \mathcal{T}} [L(h_0(x), f_Q(x)) + L(h_0(x), f_P(x))]
\tag{13}
$$

From which we deduce that,

$$
\min_{h_0 \in H_\epsilon^K} \left( \mathcal{L}_{\widehat{P}}(h_0, f) + \mathcal{L}_{\widehat{Q}_K}(h_0, f) \right) \leq \eta_{H_\epsilon^K}
\tag{14}
$$

Let's now consider $h^* \in H$, such that:

$$
h^* = \arg\min_{h \in H} \max_{x \in \mathcal{S} \cup \mathcal{T}} [L(h(x), f_Q(x)) + L(h(x), f_P(x))]
\tag{15}
$$

By assumption we have $\eta_H \leq \epsilon$, and thus:

$$
\max_{x \in \mathcal{S} \cup \mathcal{T}} [L(h^*(x), f_Q(x)) + L(h^*(x), f_P(x))] \leq \epsilon
\tag{16}
$$

This implies that for any $x \in \mathcal{S} \cup \mathcal{T}$:

$$
L(h^*(x), f_Q(x)) \leq \epsilon
\tag{17}
$$

In particular, $L(h^*(x), f_Q(x)) \leq \epsilon$ for any $x \in \mathscr{L}_K$, which implies that $h^*$ is in $H_\epsilon^K$. We then deduce that:

$$
\eta_H = \eta_{H_\epsilon^K}
\tag{18}
$$

Thus we conclude that for any $h \in H_\epsilon^K$ and any $\delta > 0$, we have with probability at least $1 - \delta$:

$$
\mathcal{L}_P(h, f_P) \leq \mathcal{L}_{\widehat{Q}_K}(h, f_Q) + \mathrm{disc}_{H_\epsilon^K}(\widehat{Q}_K, \widehat{P}) + \eta_H + 2\mu \mathfrak{R}_n(H) + M \left( \sqrt{\frac{\log(\frac{1}{\delta})}{2n}} \right).
\tag{19}
$$

$\square$

## B PROOF OF THEOREM 1

**Theorem 1.** Let $K > 0$ be the number of queries, $H$ a hypothesis space of $k$-Lipschitz functions and $\epsilon \geq \eta_H$. Let $\mathscr{L}_K = \mathcal{S} \cup \mathcal{T}_K$ be the labeled set and $\mathcal{T}$ the target set drawn according to $P$. We assume that $L$ is a symmetric, $\mu$-Lipschitz and bounded loss function verifying the triangular inequality. We denote by $M$ the bound of $L$. For any hypothesis $h \in H_\epsilon^K$ and any $\delta > 0$, the following generalization bound holds with at least probability 1-$\delta$:

$$\mathcal{L}_P(h, f) \leq \mathcal{L}_{\widehat{Q}_K}(h, f_Q) + \frac{2k\mu}{n} \sum_{x' \in \mathcal{T}} d(x', \mathscr{L}_K) + 2\epsilon + \eta_H + 2\mu \mathfrak{R}_n(H) + \sqrt{\frac{M^2 \log(\frac{1}{\delta})}{2n}} \quad (20)$$

With $d(x', \mathscr{L}_K) = \min_{x \in \mathscr{L}_K} d(x', x)$

*Proof.* Let $\epsilon \geq \eta_H$ and $0 \leq K \leq n$.

For all $h, h' \in H_\epsilon^K$, for all $x' \in \mathcal{T}$ and for all $x \in \mathscr{L}_K$ we have:

$$
\begin{aligned}
L(h(x'), h'(x')) &\leq L(h(x'), h(x)) + L(h(x), h'(x)) + L(h'(x), h'(x')) \\
&\leq L(h(x'), h(x)) + L(h'(x), h'(x')) + L(h(x), f_Q(x)) + L(f_Q(x), h'(x)) \\
&\leq \mu \left( |h(x) - h(x')| + |h'(x) - h'(x')| \right) + 2\epsilon \\
&\leq 2k\mu \, d(x', x) + 2\epsilon
\end{aligned}
\quad (21)
$$

The two first inequalities come from the triangular inequality, the others from the lipschitzness of $h, h'$ and definition of $H_\epsilon^K$.

As the above inequality is true for any $x \in \mathscr{L}_K$, we have in particular for any $x' \in \mathcal{T}$:

$$
\begin{aligned}
L(h(x'), h'(x')) &\leq 2k\mu \min_{x \in \mathscr{L}_K} d(x, x') + 2\epsilon \\
&= 2k\mu \, d(x', \mathscr{L}_K) + 2\epsilon
\end{aligned}
\quad (22)
$$

Leading to:

$$\mathcal{L}_{\widehat{P}}(h(x'), h'(x')) \leq \frac{2k\mu}{n} \sum_{x' \in \mathcal{T}} d(x', \mathscr{L}_K) + 2\epsilon \quad (23)$$

We then deduce the following, for all $h, h' \in H_\epsilon^K$:

$$
\begin{aligned}
\operatorname{disc}_{H_\epsilon^K}(\widehat{Q}_K, \widehat{P}) &= \max_{h, h' \in H_\epsilon^K} |\mathcal{L}_{\widehat{P}}(h, h') - \mathcal{L}_{\widehat{Q}_K}(h, h')| \\
&\leq \max \left[ \max_{h, h' \in H_\epsilon^K} \mathcal{L}_{\widehat{P}}(h, h'), \max_{h, h' \in H_\epsilon^K} \mathcal{L}_{\widehat{Q}_K}(h, h') \right] \\
&\leq \max \left[ \frac{2k\mu}{n} \sum_{x' \in \mathcal{T}} d(x', \mathscr{L}_K) + 2\epsilon, 2\epsilon \right] \\
&\leq \frac{2k\mu}{n} \sum_{x' \in \mathcal{T}} d(x', \mathscr{L}_K) + 2\epsilon
\end{aligned}
\quad (24)
$$

Finally, according to Proposition 1, we have for all $h, h' \in H_\epsilon^K$:

$$\mathcal{L}_P(h, f) \leq \mathcal{L}_{\widehat{Q}_K}(h, f_Q) + \frac{2k\mu}{n} \sum_{x' \in \mathcal{T}} d(x', \mathscr{L}_K) + 2\epsilon + \eta_H + 2\mu \mathfrak{R}_n(H) + \sqrt{\frac{M^2 \log(\frac{1}{\delta})}{2n}} \quad (25)$$

$\square$

## C   APPROXIMATION ERROR BETWEEN PROPOSITION 1 AND THEOREM 1

We present in this section the approximation error of the relaxation between the bounds of Proposition 1 and Theorem 1.

We will show that, with the assumption of Theorem 1 and in the case $L = L_1$, we have, for any labeled set $\mathscr{L}_K$:

$$\frac{2k}{n} \sum_{x' \in \mathcal{T}} d(x', \mathscr{L}_K) \leq \frac{k}{k - k_f} \text{disc}_{H_\epsilon^K}(\widehat{Q}_K, \widehat{P}) \tag{26}$$

With $k_f$ the Lipschitz constant of the source labeling function $f_Q$. We assume that $k > k_f$.

For this purpose we will look for two hypotheses $h, h' \in H_\epsilon^K$ verifying:

$$\mathcal{L}_{\widehat{P}}(h, h') \geq \frac{2(k - k_f)}{n} \sum_{x' \in \mathcal{T}} d(x', \mathscr{L}_K) \tag{27}$$

Let's define $k' = k - k_f$, and $h, h' : \mathcal{X} \to \mathcal{Y}$ such that, for any $x \in \mathcal{X}$:

$$
\begin{aligned}
h(x) &= f_Q(x) + k' \min_{\tilde{x} \in \mathscr{L}_K} d(x, \tilde{x}) = f_Q(x) + k' d(x, \mathscr{L}_K) \\
h'(x) &= f_Q(x) - k' \min_{\tilde{x} \in \mathscr{L}_K} d(x, \tilde{x}) = f_Q(x) - k' d(x, \mathscr{L}_K)
\end{aligned}
\tag{28}
$$

We will now prove that $h, h'$ are in $H_\epsilon^K$:

Let's consider $x_1, x_2 \in \mathcal{X}$, we define:

$$
\begin{aligned}
\tilde{x}_1 &= \underset{\tilde{x} \in \mathscr{L}_K}{\arg \min}\, d(\tilde{x}, x_1) \\
\tilde{x}_2 &= \underset{\tilde{x} \in \mathscr{L}_K}{\arg \min}\, d(\tilde{x}, x_2)
\end{aligned}
\tag{29}
$$

Assuming without restriction that $d(x_1, \tilde{x}_1) > d(x_2, \tilde{x}_2)$, we have:

$$
\begin{aligned}
|h(x_1) - h(x_2)| &\leq k' \left| d(x_1, \mathscr{L}_K) - d(x_2, \mathscr{L}_K) \right| + k_f\, d(x_1, x_2) \\
&\leq k' \left( d(x_1, \tilde{x}_1) - d(x_2, \tilde{x}_2) \right) + k_f\, d(x_1, x_2) \\
&\leq k' \left( d(x_1, \tilde{x}_2) - d(x_2, \tilde{x}_2) \right) + k_f\, d(x_1, x_2) \\
&\leq k'\, d(x_1, x_2) + k_f\, d(x_1, x_2) \\
&\leq k\, d(x_1, x_2)
\end{aligned}
\tag{30}
$$

Using the triangular inequality and the fact that $d(x_1, \tilde{x}_1) \leq d(x_1, \tilde{x}_2)$ by definition of $\tilde{x}_1$.

Using a similar development we can prove the $k$-Lipschitzness of $h'$.

Let's now consider $x \in \mathscr{L}_K$, we have

$$
\begin{aligned}
L(h(x), f_Q(x)) &= L(f_Q(x) + k' d(x, \mathscr{L}_K), f_Q(x)) \\
&= L(f_Q(x), f_Q(x)) \\
&= 0
\end{aligned}
\tag{31}
$$

In the same way $L(h'(x), f_Q(x)) = 0$ and we have $h, h' \in H_\epsilon^K$.

Furthermore, we have:

$$
\begin{aligned}
\mathcal{L}_{\widehat{P}}(h, h') &= \frac{1}{n} \sum_{x' \in \mathcal{T}} L(h(x'), h'(x')) \\
&= \frac{1}{n} \sum_{x' \in \mathcal{T}} |h(x') - h'(x')| \\
&= \frac{2(k - k_f)}{n} \sum_{x' \in \mathcal{T}} d(x', \mathscr{L}_K)
\end{aligned}
\tag{32}
$$

Thus,

$$
\mathrm{disc}_{H_\epsilon^K}(\widehat{Q}_K, \widehat{P}) \geq |\mathcal{L}_{\widehat{P}}(h, h') - 0| = \frac{2(k - k_f)}{n} \sum_{x' \in \mathcal{T}} d(x', \mathscr{L}_K)
\tag{33}
$$

From which we conclude.

## D    COMPARISON WITH OTHER ACTIVE LEARNING BOUNDS (CF SECTION 4.2)

In this section we assume that $f = f_P = f_Q$ and $f \in H$ with $H$ a set of $k$-Lipschitz functions.

### D.1    K-CENTER BOUNDS

Sener and Savarese (Sener & Savarese, 2018) propose the following generalization bounds for $\epsilon = 0$ and $h \in H_\epsilon^K$:

$$
\mathcal{L}_{\widehat{P}}(h, f) \leq \delta(\lambda^l + MC\lambda^\mu) + M\left(\sqrt{\frac{\log(\frac{1}{\delta})}{2n}}\right)
\tag{34}
$$

which leads to:

$$
\mathcal{L}_P(h, f) \leq \delta(\lambda^l + MC\lambda^\mu) + 2\mu\mathfrak{R}_n(H) + 2M\left(\sqrt{\frac{\log(\frac{1}{\delta})}{2n}}\right)
\tag{35}
$$

With $\delta = \max_{x' \in \mathcal{T}} d(x', \mathscr{L}_K)$, $C$ the class number and $\lambda^\mu$ the Lipschitz constant of a class-specific regression function. $\lambda^l$ is the Lipschitz constant of the loss function $l$ verifying $l : (x, h) \rightarrow l(x, f(x), h) = L(h(x), f(x))$.

If we consider a regression problem, we can drop the term corresponding to the class-specific function and we have:

$$
\mathcal{L}_P(h, f) \leq \delta\lambda^l + 2\mu\mathfrak{R}_n(H) + \mathcal{O}\left(\sqrt{\frac{M^2 \log(\frac{1}{\delta})}{2n}}\right)
\tag{36}
$$

Let's now consider $h \in H$ and $x, x' \in \mathcal{X}$, we have for $k$-Lipschitz $f$ :

$$
\begin{aligned}
|l(x, f(x), h) - l(x', f(x'), h)| &= |L(h(x), f(x)) - L(h(x'), f(x'))| \\
&\leq |L(h(x), f(x)) - L(h(x'), f(x))| + |L(h(x'), f(x)) - L(h(x'), f(x'))| \\
&\leq L(h(x), h(x')) + L(f(x), f(x')) \\
&\leq \mu|h(x) - h(x')| + \mu|f(x) - f(x')| \\
&\leq 2\mu k|x - x'|
\end{aligned}
\tag{37}
$$

The two first inequalities are obtained with triangular inequalities, then we use Lipschitz assumptions on $h$, $f$ and $L$.

Thus, we have $\lambda^l = 2\mu k$ from which we deduce:

$$
\mathcal{L}_P(h, f) \leq 2k\mu \max_{x' \in \mathcal{T}} d(x', \mathscr{L}_K) + 2\mu\mathfrak{R}_n(H) + \mathcal{O}\left(\sqrt{\frac{M^2 \log(\frac{1}{\delta})}{2n}}\right)
\tag{38}
$$

### D.2 WASSERSTEIN BOUNDS

To adapt the bound from (Shui et al., 2020) Corollary 1 to our setting, we identify the distributions $\widehat{D}$ and $\widehat{Q}$ from Shui et al. (2020) with respectively the distributions $\widehat{P}$ and $\widehat{Q}_K$. We then have the following generalization bound:

$$
\mathcal{L}_P(h, f) \leq \mathcal{L}_{\widehat{Q}_K}(h, f) + 2\mu k W_1(\widehat{Q}_K, \widehat{P}) + 2\mu\mathfrak{R}_n(H) + \mathcal{O}\left(\sqrt{\frac{M^2 \log(\frac{1}{\delta})}{2n}}\right)
\tag{39}
$$

Notice that the term corresponding to the labeling function "decay property" in the bound of Shui et al. (2020) is null when considering a Lipschitz labeling function ($f \in H$).

Thus for $\epsilon = 0$ and for any $h \in H_\epsilon^K$, we have:

$$
\begin{aligned}
\mathcal{L}_P(h, f) &\leq 2k\mu W_1(\widehat{Q}_K, \widehat{P}) + 2\mu\mathfrak{R}_n(H) + \mathcal{O}\left(\sqrt{\frac{M^2 \log(\frac{1}{\delta})}{2n}}\right) \\
&= 2k\mu \sum_{x' \in \mathcal{T}} \sum_{x \in \mathscr{L}_K} \gamma_{x'x}^* d(x', x) + 2\mu\mathfrak{R}_n(H) + \mathcal{O}\left(\sqrt{\frac{M^2 \log(\frac{1}{\delta})}{2n}}\right)
\end{aligned}
\tag{40}
$$

With $\gamma^* = \arg\min_{\gamma \in \Gamma} \sum_{x' \in \mathcal{T}} \sum_{x \in \mathscr{L}_K} \gamma_{x'x} d(x', x)$ and $\Gamma = \{\gamma \in \mathbb{R}^{n \times (m+K)} \, ; \, \gamma\mathbf{1} = \frac{1}{n}\mathbf{1} \, ; \, \gamma^T\mathbf{1} = \frac{1}{m+K}\mathbf{1}\}$.

Thus, in observing that:

$$
\begin{aligned}
\sum_{x' \in \mathcal{T}} \sum_{x \in \mathscr{L}_K} \gamma_{x'x} d(x', x) &\geq \sum_{x' \in \mathcal{T}} \sum_{x \in \mathscr{L}_K} \gamma_{x'x} \min_{x \in \mathscr{L}_K} d(x', x) \\
&\geq \sum_{x' \in \mathcal{T}} d(x', \mathscr{L}_K) \sum_{x \in \mathscr{L}_K} \gamma_{x'x} \\
&\geq \frac{1}{n} \sum_{x' \in \mathcal{T}} d(x', \mathscr{L}_K)
\end{aligned}
$$

We deduce that our presented bound of Theorem 1 is also tighter than the one proposed in (Shui et al., 2020) in the case of $\epsilon = 0$ and $k$-Lipschitz $f$ and $h$.

# E  ALGORITHMS (CF SECTION 3)

## E.1  K-MEDOIDS

---
**Algorithm 2** K-Medoids Greedy

---
1: **Input:** $\mathcal{T}$, $B$, $K$, $\mathcal{S}$, $\{d(x', \mathcal{S})\}_{x' \in \mathcal{T}}$)
2: **Output:** $\mathcal{T}_K = \{x_j\}_{j \leq K} \subset \mathcal{T}$
3: $\mathcal{T} \leftarrow \{x'_{s_1}, ..., x'_{s_B}\}$ with $\{x'_{s_1}, ..., x'_{s_B}\}$ picked randomly in $\mathcal{T}$ without replacement.
4: Initialize query subset: $\mathcal{T}_0 = \{\}$
5: For all $x' \in \mathcal{T}$, $d^{x'} = d(x', \mathcal{S}) = \min_{x \in \mathcal{S}} d(x', x)$
6: For all $x, x' \in \mathcal{T}$ compute $d^{xx'} = d(x, x')$
7: **for** $i$ from 1 to $K$ **do**
8:   $x_i \leftarrow \arg\min_{x \in \mathcal{T}} \sum_{x' \in \mathcal{T}} \min\left(d^{xx'}, d^{x'}\right)$
9:   $\mathcal{S}_i \leftarrow \mathcal{S}_{i-1} \cup \{x_i\}$
10:   For all $x' \in \mathcal{T}$, update $d^{x'} = \min(d^{x'}, d^{x_i x'}))$
11: **end for**

---

---
**Algorithm 3** Branch & Bound Medoid (B & B)

---
1: **Input:** Cluster $C \in \mathbb{R}^{n_c \times p}$, previous medoid criterion $\mathcal{C}^*$, batch size $B$
2: **Output:** New medoid $x^*$
3: Initialize candidates $\tilde{C} = C$
4: Initialize computed distance set $D_x = \{\}$ for all $x \in \tilde{C}$.
5: Initialize criterion $\mathcal{C}_x = 0$ and standard deviation $\sigma_x = 0$ for all $x \in \tilde{C}$.
6: Initialize threshold $t = \mathcal{C}^*$
7: **for** $i$ from 1 to $n_c/B$ **do**
8:   $C_i = \{x_j \in C \ ; \ (B-1)i \leq j \leq Bi\}$
9:   **for** $x \in \tilde{C}$ **do**
10:     $\mathcal{D}_x \leftarrow \mathcal{D}_x \cup \{d(x, x') \ ; \ x' \in C_i\}$
11:     $\mathcal{C}_x \leftarrow \frac{1}{Bi} \sum_{d \in \mathcal{D}_x} d$
12:     $\sigma_x \leftarrow \sqrt{\frac{1}{Bi} \sum_{d \in \mathcal{D}_x} (d - \mathcal{C}_x)^2}$
13:   **end for**
14:   $x^* \leftarrow \arg\min_{x \in \tilde{C}} \mathcal{C}_x$
15:   $t \leftarrow \min(t, \mathcal{C}_{x^*} + \frac{2\sigma_{x^*}}{\sqrt{Bi}})$
16:   $\tilde{C} \leftarrow \left\{x \in \tilde{C} \ ; \ \mathcal{C}_x - \frac{2\sigma_x}{\sqrt{Bi}} < t\right\}$
17: **end for**
18: $x^* \leftarrow \arg\min_{x \in \tilde{C}} \mathcal{C}_x$

---

## E.2  APPROXIMATION BOUND FOR THE GREEDY ALGORITHM

This section is dedicated to the proof of the bound for the greedy K-medoids algorithm which is expressed as follows:

*The gain of selecting $K$ new medoids with the greedy algorithm is an $(1 - 1/e)$-approximation of the optimal gain.*

Let $\mathcal{T}_K$ be a batch of $K$ target points selected with the greedy algorithm. The gain is the difference between the initial objective and the final objective after selecting $\mathcal{T}_K$:

$$\text{gain}(\mathcal{T}_K) = \sum_{x' \in \mathcal{T}} d(x', \mathcal{S}) - \sum_{x' \in \mathcal{T}} d(x', \mathcal{S} \cup \mathcal{T}_K) \tag{41}$$

With $d(x', \mathcal{S}) = \min_{x \in \mathcal{S}} d(x', x)$

We will prove now that the gain is monotone submodular.

Let $A \subset B \subset \mathcal{T}$ and $x \in \mathcal{T} \setminus B$,

We denote $\mathcal{T}^A = \{x' \in \mathcal{T}; d(x', x) < d(x', A)\}$ and $\mathcal{T}^B = \{x' \in \mathcal{T}; d(x', x) < d(x', B)\}$.

As $A \subset B$, we have $d(x', A) \geq d(x', B)$ for any $x' \in \mathcal{T}$. Thus,

$$\mathcal{T}^B \subset \mathcal{T}^A \tag{42}$$

Besides, for any $x' \in \mathcal{T}$,

$$d(x', A) - d(x', x) \geq d(x', B) - d(x', x) \tag{43}$$

We deduce then that,

$$
\begin{aligned}
\mathrm{gain}(A \cup \{x\}) - \mathrm{gain}(A) &= \sum_{x' \in \mathcal{T}^A} d(x', A) - d(x', x) \\
&\geq \sum_{x' \in \mathcal{T}^B} d(x', A) - d(x', x) \\
&\geq \sum_{x' \in \mathcal{T}^B} d(x', B) - d(x', x) \\
&= \mathrm{gain}(B \cup \{x\}) - \mathrm{gain}(B)
\end{aligned}
\tag{44}
$$

Then, the gain is submodular.

Besides, as $A \subset B$ it appears clearly that,

$$\mathrm{gain}(A) \subset \mathrm{gain}(B) \tag{45}$$

and the gain is monotone.

Finally, by noticing that the gain is always positive and according to (Nemhauser et al., 1978), we conclude that the gain of selecting $K$ new medoids with the greedy algorithm is an $(1 - 1/e)$-approximation of the optimal gain.

### E.3 COMPLEXITY COMPUTATION

In the following, a distance computation is considered to be done in $\mathcal{O}(p)$.

1. **KD-Trees Random Forest**: Each of the $T$ trees is built by splitting one sample, at each root, at the median of a random feature until the leaf-sizes are $\sim \log(m)$. The median computation for each root with $m_r$ data is in $\mathcal{O}(m_r \log(m_r))$. Thus the overall complexity to build one tree is $\mathcal{O}(\sum_{i=0}^{M} 2^{-i} m \log(2^{-i} m) 2^i) = \mathcal{O}(m \log(m) \sum_{i=0}^{M} 1)$ with $M \sim \log(m/\log(m))$ which becomes $\mathcal{O}(m \log(m)^2)$. Then, each target is assigned to a leaf by performing $\mathcal{O}(\log(m))$ computations, inside the assigned leaf all distance computations are done in $\mathcal{O}(p \log(m))$. Thus an approximate nearest neighbour is given for all targets with a complexity $\mathcal{O}(T(m + pn) \log(m)^2)$.

2. **Medoids Initialization**: Using the greedy algorithm, $\mathcal{O}(pB^2)$ distance computations are first done, then, for all targets, a sum over the target data is computed at each of the $K$ steps. Thus the complexity is $\mathcal{O}((K + p)B^2)$.

3. **Assignation to the Closest Medoid**: The distance between all target and the medoids is computed in $\mathcal{O}(Kpn)$.

4. **Branch & Bound Medoid Computation**

   B & B algorithm (Algorithm 3) takes as input one cluster $C$ of $n_c$ unlabeled data from $\mathcal{T}$. It also takes a batch size $B$ and the previous cluster medoid criterion $\mathcal{C}^*$ which is used as

an initial threshold. The use of the initialization $\mathcal{C}^*$ may accelerate the algorithm, in the following we do not take into account this initialization, i.e we consider that $\mathcal{C}^* = +\infty$.

Besides, to use notations consistent with the common notations in statistics, we will denote the batch size $B$ by $n$ ($B \equiv n$). Notice that it is redundant with the size of the unlabeled data set $\mathcal{T}$. An explicit mention will be made, if $n$ does not refer to the batch size.

**Definitions and notations:** Let's consider one cluster $C \subset \mathcal{T}$ with $n_c$ data. We consider the uniform norm as underlying distance $d$, defined for all $x_i, x_j \in C$ as $d(x_i, x_j) = \max\left(|x_i^{(1)} - x_j^{(1)}|, ..., |x_i^{(p)} - x_j^{(p)}|\right)$ with $x_i = (x_i^{(1)}, ..., x_i^{(p)}) \in \mathbb{R}^p$.

Computing the complexity of the B & B algorithm for any distribution of the $x_i$ would be too difficult. We make here the simplifying assumption that the $x_i$ in $C$ are uniformly distributed on the hyper-cube $C$ of edge size 2 and centered on $x^* = (0, ..., 0) \in \mathbb{R}^p$

We define for any $i \in [|1, n_c|]$ and any $j \in [|1, n_c|]$, the variables $Z_j^i = d(x_i, x_j)$. We also define $Z_j^* = d(x^*, x_j)$ for any $j \in [|1, n_c|]$. We suppose that for any $i \in [|1, n_c|]$, $Z_1^i, ..., Z_{n_c}^i$ are iid and that for any $j \in [|1, n_c|]$ the $Z_j^i$ are independents. We define, for any $i \in [|1, n_c|]$, the mean $\mu_i = \mathrm{E}[Z_0^i] = \frac{1}{2^p} \int_{x \in C} d(x_i, x)$ and the variance $\sigma_i^2 = \mathrm{Var}[Z_0^i] = \frac{1}{2^p} \int_{x \in C} (d(x_i, x) - \mu_i)^2$. We consider a first batch of distance computations of size $n < n_C$. We define for any $i \in [|1, n_c|]$ the empirical mean $\widehat{\mu}_i = \frac{1}{n} \sum_{j=1}^{n} Z_j^i$ and the empirical variance $\widehat{\sigma}_i^2 = \frac{1}{n-1} \sum_{j=1}^{n} (Z_j^i - \widehat{\mu}_i)^2$. We denote by $\mu^*$ and $\sigma^{*2}$ the mean and variance of $Z_0^*$ and $\widehat{\mu}^*$ and $\widehat{\sigma}^{*2}$ their respective empirical estimator.

We first observe that $\mu_i$ and $\sigma_i$ are finite for any $i \in [|1, n_c|]$ as the $x_i$ are uniformly distributed on the hyper-cube centered in $x^* \in C$. We also notice that $Z_j^i \in [0, 2]$ for any $i, j \in [|1, n_c|]$.

**Preliminary results:** We make the assumption that $p >> 1$. We aim at giving bounds for any $\mu_i$ and $\sigma_i$. We admit the intuitive results that for any $x_i = (x_i^{(1)}, ..., x_i^{(p)}) \in C$ and for any $j \in [|1, n_c|]$, $\mu_i \geq \mu_i^{j \to 0}$ and $\sigma_i \geq \sigma_i^{j \to 0}$ with $\mu_i^{j \to 0}$ and $\sigma_i^{j \to 0}$ the mean and variance of the variable $d(x_i^{j \to 0}, .)$ with $x_i^{j \to 0} = (x_i^{(1)}, ..., x_i^{(j-1)}, 0, x_i^{(j+1)}, ..., x_i^{(p)})$. We consider indeed that the more $x_i$ is close to the center of the hyper-cube smaller is $\mu_i$ and $\sigma_i$. Considering this fact, we have, for any $i \in [|1, n_c|]$:

$$\mu^* \leq \mu_i \tag{46}$$

$$\sigma^* \leq \sigma_i \tag{47}$$

Besides, for any $0 \leq r \leq 1$ the sample density in the elementary surface between the balls centered on $x^*$ and of respective radius $r + \mathrm{d}r$ and $r$ is $pr^{p-1}\mathrm{d}r$. Thus, we can notice that $Z_0^*$ follows a beta distribution of parameters $\alpha = p$ and $\beta = 1$, from which we deduce that:

$$\mu^* = \frac{p}{p+1} \tag{48}$$

$$\sigma^{*2} = \frac{p}{(p+1)^2(p+2)} \tag{49}$$

We further notice that for any $0 \leq a \leq 1$ and for any $i \in [|1, n_c|]$ such that $d(x^*, x_i) = a$ we have $\mu_i \geq \mu_a$ with $\mu_a$ the criterion of the sample $x_a = (a, 0, ..., 0) \in \mathbb{R}^p$.

To compute $\mu_a = \frac{1}{2^p} \int_{x \in C} d(x_a, x)$, we split the integral on three parts: $d(x_a, x) \leq 1 - a$, $1 - a \leq d(x_a, x) \leq 1$ and $1 \leq d(x_a, x) \leq 1 + a$:

$$
\begin{aligned}
\mu_a &= \int_{r=0}^{1-a} pr^{p-1}\mathrm{d}r + \frac{1}{2}\int_{r=1-a}^{1}\left(pr^{p-1} + (p-1)(1-a)r^{p-2}\right)\mathrm{d}r + \frac{1}{2}\int_{1}^{1+a} r\mathrm{d}r \\
&= \frac{p}{p+1}(1-a)^{p+1} + \frac{1}{2}\left[\frac{p}{p+1} + (1-a)\frac{p-1}{p} - (1-a)^{p+1}\left(\frac{p}{p+1} + \frac{p-1}{p}\right)\right] + \frac{a}{2}\left(1 + \frac{a}{2}\right) \\
&\simeq 1 - \frac{a}{2} + \frac{a}{2}\left(1 + \frac{a}{2}\right) \\
&\simeq 1 + \frac{a^2}{4}
\end{aligned}
$$

$$\tag{50}$$

Using the simplifying approximation $\frac{p}{p+1} \simeq \frac{p-1}{p} \simeq 1$ for $p >> 1$. Thus for any $0 \le a \le 1$ and for any $i \in [|1, n_c|]$ such that $d(x^*, x_i) = a$ we have:

$$\mu_i \ge 1 + \frac{a^2}{4} \tag{51}$$

An upper bound of the $\sigma_i^2$ is given by the variance $\sigma_c^2$ of the variable $d(x_c, .)$ with $x_c = (1, ..., 1)$ which corresponds to one corner of the hyper-cube $C$:

$$
\begin{aligned}
\sigma_c^2 &= \frac{1}{2^p} \int_0^2 pr^{p+1} \mathrm{d}r - \left( \frac{1}{2^p} \int_0^2 pr^p \mathrm{d}r \right)^2 \\
&= 4\frac{p}{p+2} + 4\left( \frac{p}{p+1} \right)^2 \\
&= 4\frac{p}{(p+1)^2(p+2)} \\
&= 4\sigma^{*2}
\end{aligned}
\tag{52}
$$

From which we deduce that for any $i \in [|1, n_c|]$:

$$\sigma^* \le \sigma_i \le 2\sigma^* \tag{53}$$

To simplify the calculations, we make the approximation $\widehat{\sigma}_i \simeq \sigma_i$ for any $i \in [|1, n_c|]$, which is relevant for sufficiently large $n$ as $\mathrm{E}[Z_0^{i\,4}] < +\infty$.

**Probability of rejecting all optimal medoid candidates:** Let $\epsilon > 0$ be an approximation factor of $\mu^*$. The goal of the Branch & Bound algorithm is to identify one sample $x_i \in C$ such that $\mu_i \le \mu^*(1 + \epsilon)$ with less distance computations as possible. The process consists in removing all candidates $x_i$ such that $\widehat{\mu}_i - \frac{2\widehat{\sigma}_i}{\sqrt{n}} > \widehat{\mu}_{i^*} + \frac{2\widehat{\sigma}_{i^*}}{\sqrt{n}}$ with $\widehat{\mu}_{i^*}$ the current minimal empirical mean.

We aim now at computing the probability of rejecting all optimal medoid candidates $x_i$ verifying $\mu_i \le \mu^*(1 + \epsilon)$ during the B & B process. For this, we define $\mathcal{B}_\epsilon(\mu^*) = \{i \in [|1, n_c|]; \mu_i \le \mu^*(1 + \epsilon)\}$ the index set of optimal medoid candidates and $\mathcal{B}_\epsilon^c(\mu^*) = \{i \in [|1, n_c|]; \mu_i > \mu^*(1 + \epsilon)\}$ the index set of sub-optimal medoid candidates. We assume that B & B returns an optimal candidate if at least one sample of $\mathcal{B}_\epsilon(\mu^*)$ is kept after the first batch computation. We define the two following probabilities:

$$P_1 = P\left( \left\{ \exists i \in \mathcal{B}_\epsilon^c(\mu^*); \widehat{\mu}_i + \frac{4\sigma^*}{\sqrt{n}} \le \mu^*(1 + \epsilon/4) \right\} \right) \tag{54}$$

$$P_2 = P\left( \{ \exists i \in \mathcal{B}_\epsilon(\mu^*); \widehat{\mu}_i \le \mu^*(1 + \epsilon/4) \} \right) \tag{55}$$

We can observe that the probability of rejecting all optimal medoid candidates is upper bounded by: $(1 - P_2) + P_2 P_1$ considering the approximation $\widehat{\sigma}_i \simeq \sigma_i$ and the fact that $\sigma_i \ge \sigma^*$ for any $i \in [|1, n_c|]$.

We now define, for $i \in \mathcal{B}_\epsilon^c(\mu^*)$:

$$P_i = P\left( \left\{ \widehat{\mu}_i + \frac{4\sigma^*}{\sqrt{n}} \le \mu^*(1 + \epsilon/4) \right\} \right) \tag{56}$$

Leading to:

$$P_i = P\left( \left\{ \widehat{\mu}_i + \frac{4\sigma^*}{\sqrt{n}} - \frac{\epsilon\mu^*}{4} \le \mu^* \right\} \right) \tag{57}$$

On the other hand, according to the Bennett's inequality from (Maurer & Pontil, 2009; Hoeffding, 1994) we have, for any $i \in \mathcal{B}_\epsilon^c(\mu^*)$ and for any $\delta > 0$:

$$\mathrm{P}\left( \widehat{\mu}_i/2 + \sqrt{\frac{\sigma_i^2 \log(1/\delta)}{2n}} + \frac{\log(1/\delta)}{3n} \le \mu_i/2 \right) \le \delta \tag{58}$$

Notice that we apply the inequality to the $Z_j^i/2$. Then, considering the fact that $\sigma_i \leq 2\sigma^*$ and that $\mu_i > \mu^*(1 + \epsilon)$ we have:

$$\mathrm{P}\left(\widehat{\mu}_i + \frac{\sqrt{2}\sigma^*}{\sqrt{n}}\sqrt{\log(1/\delta)} + \frac{2\log(1/\delta)}{3n} \leq \mu^*(1 + \epsilon)\right) \leq \delta \tag{59}$$

Leading to:

$$\mathrm{P}\left(\widehat{\mu}_i + \frac{\sqrt{2}\sigma^*}{\sqrt{n}}\sqrt{\log(1/\delta)} - \epsilon\mu^* + \frac{2\log(1/\delta)}{3n} \leq \mu^*\right) \leq \delta \tag{60}$$

Let's consider $\delta > 0$ such that the following equality holds:

$$\frac{\sqrt{2}\sigma^*}{\sqrt{n}}\sqrt{\log(1/\delta)} - \epsilon\mu^* + \frac{2\log(1/\delta)}{3n} = \frac{4\sigma^*}{\sqrt{n}} - \frac{\epsilon\mu^*}{4} \tag{61}$$

Thus:

$$u^2 + Au = B \tag{62}$$

With:

$$u = \sqrt{\log(1/\delta)} \tag{63}$$

$$A = \frac{3}{2}\sqrt{2n}\sigma^* \tag{64}$$

$$B = \frac{3n}{2}\left(\frac{4\sigma^*}{\sqrt{n}} + \frac{3\epsilon\mu^*}{4}\right) \tag{65}$$

$$\tag{66}$$

We then set:

$$\delta = \exp\left(-\left(\frac{\Delta - A}{2}\right)^2\right) \tag{67}$$

$$\Delta = \sqrt{A^2 + 4B} \tag{68}$$

We have for the $\delta$ defines above and for any $i \in \mathcal{B}_\epsilon^c(\mu^*)$:

$$P_i \leq \delta \tag{69}$$

Thus, the probability $P_1$ can be upper bounded as follows:

$$P_1 \leq 1 - (1 - \delta)^{n_c} \tag{70}$$

Considering the fact that $|\mathcal{B}_\epsilon^c(\mu^*)| < n_c$. $1 - (1 - \delta)^{n_c}$ is the probability of getting at least one success for the binomial law of parameters $(n_c, \delta)$.

We are now looking for an upper bound of $P_2$. We observe that, at least the index $i$ such that $x_i = x^*$ is in $\mathcal{B}_\epsilon(\mu^*)$. Besides, as $x^*$ is the center of the hyper-cube, $Z_j^*$ is in $[0, 1]$ for any $j \in [|1, n_c|]$. We can then apply the Bennett's inequality to the $Z_j^*$, and for any $\gamma > 0$ we have:

$$\mathrm{P}\left(\widehat{\mu}^* \leq \mu^* + \sqrt{\frac{2\sigma^{*2}\log(1/\delta)}{n}} + \frac{\log(1/\delta)}{3n}\right) \geq 1 - \gamma \tag{71}$$

We set:

$$\gamma = \exp\left(-\left(\frac{\sqrt{C^2 + 4D} - C}{2}\right)^2\right) \tag{72}$$

$$C = 3\sqrt{2n}\sigma^* \tag{73}$$

$$D = 3n\frac{\epsilon\mu^*}{4} \tag{74}$$

We then have:

$$P_2 \geq \mathrm{P}\left(\widehat{\mu}^* \leq \mu^*(1 + \frac{\epsilon}{4})\right) \geq 1 - \gamma \tag{75}$$

Leading to:

$$1 - P_2 \leq \gamma \tag{76}$$

Finally the probability of rejecting all optimal candidates is upper bounded by $1 - (1 - \delta)^{n_c} + \gamma$. To give an order of magnitude of this probability, we consider the scenario where $n_c = 10^5$, $p = 100$, $n = \sqrt{n_c}$ and $\epsilon = 0.05$. Then we have: $\delta \simeq 3.6\,10^{-8}$ and $\gamma \simeq 7.7\,10^{-5}$, which leads to a probability of rejection around $3.6\,10^{-3}$. Thus, in this case, there is at least a probability $0.995$ that B & B returns a medoid candidate with a criterion less than $1.05$ the optimal.

**Complexity computation:** We are now looking for the number of distance computations performed by B & B. For this, we need to compute the number of $x_i$ kept at each batch. An upper bound of this number is given by the number of $x_i$ verifying $\widehat{\mu}_i \leq \mu^*(1 + \epsilon/4) + \frac{8\sigma^*}{\sqrt{n}}$. We further assume that the previous upper bound can be approximated by the number of $x_i$ verifying $\mu_i \leq \mu^*(1 + \epsilon/4) + \frac{8\sigma^*}{\sqrt{n}}$

We have shown in the preliminary results that for any $i \in [|1, n_c|]$, $\mu_i \geq 1 + \frac{a^2}{4}$ with $a = d(x_i, x^*)$. Thus an upper bound of the number of candidates $x_i$ kept after the first batch is given by the number of $x_i$ in the ball of radius $a$ with $a$ verifying:

$$a = 2\sqrt{\epsilon/4 + \frac{8\sigma^*}{\sqrt{n}}} \tag{77}$$

Using the approximation $\mu^* = \frac{p}{p+1} \simeq 1$. Besides, as $\sigma^* = \mathcal{O}\left(1/p\right)$ we can suppose that for sufficiently large $p$ and large $n$, $\frac{8\sigma^*}{\sqrt{n}} \leq \frac{3}{4}\epsilon$ (For instance with $n_c, n, p$ and $\epsilon$ considered previously, we have $\frac{8\sigma^*}{\sqrt{n}} \simeq 0.005$ and $\frac{3}{4}\epsilon \simeq 0.03$). Thus:

$$a \leq 2\sqrt{\epsilon} \tag{78}$$

Finally the number of candidates kept after the first batch is in $\mathcal{O}\left(n_c\,\epsilon^{p/2}\right)$ which is very small (for the values of $n_c, p$ and $\epsilon$ considered previously, we have $n_c\,\epsilon^{p/2} \simeq 10^{-60}$). If we consider a batch size of $\mathcal{O}\left(\sqrt{n_c}\right)$, the number of distance computations after the first batch is negligible behind $\mathcal{O}\left(n_c\sqrt{n_c}\right)$.

We then conclude that the complexity of a medoid computation in one cluster is in $\mathcal{O}\left(p\,n_c\sqrt{n_c}\right)$ as each of the $K$ cluster has approximately $n_c \simeq n/K$ samples (with $n$ the sample size of $\mathcal{T}$), the overall complexity of B & B is in $\mathcal{O}\left(p\,n^{3/2}K^{-1/2}\right)$.

# F  EMPIRICAL COMPLEXITIES

This section presents the empirical time computation recorded for **K-medoids Greedy**, **Accelerated K-medoids**, **K-centers** and **K-centers + KD-Trees** which corresponds to the K-centers algorithm with initialization of the nearest source neighbour distances computed through the KD-Trees Random Forest algorithm. The experiments are conducted on the Digits data set.

The experiments are run on a (2.7GHz, 16G RAM) computer using Python 3.8. The scikit-learn (Pedregosa et al., 2011) implementation of the pairwise euclidean distance is used. No parallel computing is performed.

The parameters are set to $K = 100$, $T = 50$ for the KD-Trees Random Forest algorithm and an initial batch size $B = 5000$ for the Accelerated K-medoids algorithm. The euclidean distance is used as base distance $d$. The results are reported on Figure 5. The evolution of computational time is a function of the size of the source and target samples ($m$ and $n$).

We first observe on Figure 5 that the K-medoids Greedy algorithm encounters computational burden for samples larger than 10k instances. For $n = m = 60k$, the K-centers algorithm encounters

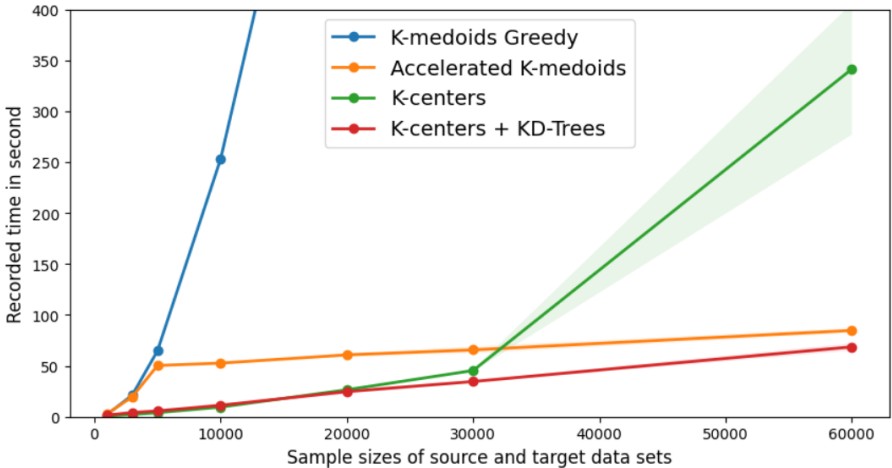

Figure 5: Visualization of empirical computational times in function of the sample sizes.

a similar issue due to the computation of the distance matrix between the source and target data sets. Using the KD-Trees Random Forest algorithm decreases in this case the computational time by a factor of $5$. We also notice that the accelerated K-medoids algorithm has similar complexity performance to K-medoids for $n, m \leq 5000$ which is the maximum size of the initial batch. Then, the computation time of the algorithm increases slightly from $n, m = 5\text{k}$ to $n, m = 60\text{k}$ while remaining at an acceptable level. For $n, m = 60\text{k}$ the complexities of the accelerated K-medoids and the K-centers + KD-Trees are almost similar but with a level of performance in favor of the accelerated K-medoids (see Section 5.4).

We also present the evolution of the objective function of the different algorithm in function of the number of queries $K$ for $n = m = 60k$ (cf Figure 6).

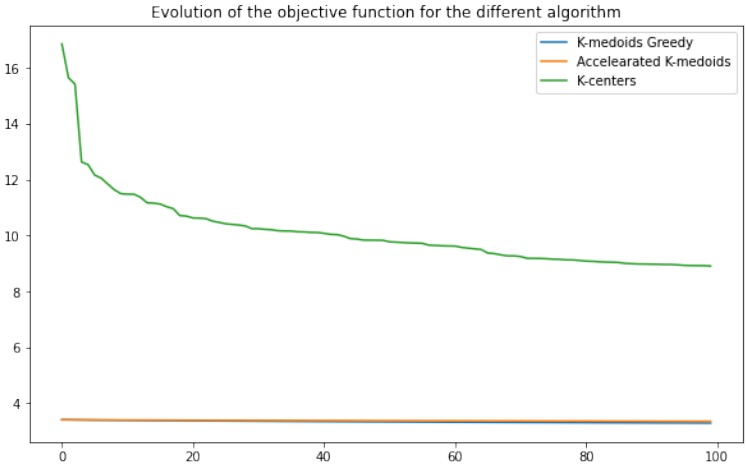

Figure 6: Visualization of empirical objectives in function of the number of queries for a sample size of 60k.

We observe that the objective of the K-center algorithm is higher than the one of the K-medoids algorithms. We also observe that K-medoids Greedy provide slightly smaller objectives than K-medoids Accelerated.

# G    EXPERIMENTS

## G.1    SUPERCONDUCTIVITY

**Setup** The UCI data set "Superconductivity" (Hamidieh, 2018; Dua & Graff, 2017) is composed of features extracted from the chemical formula of several superconductors along with their critical temperature. There is two kind of features: some features correspond to the chemical element number's in the superconductor chemical formula, others are statistical features derived from the chemical formula as the mean and variance of the atomic mass.

We use the setup of (Pardoe & Stone, 2010) to divide the data set in separate domains. We select an input feature with a moderate correlation factor with the output ($\sim 0.3$). We then sort the set according to this feature and split it in four parts: low (l), middle-low (ml), middle-high (mh), high (h). Each part defining a domain with around 4000 instances and 166 features. The considered feature is then withdrawn from the data set.

A standard scaling preprocessing is performed using the source data on the input statistical features and the output feature. A max scaling is performed on features corresponding to the chemical element number's. A visualization of the first components of the PCA as well as the output distribution is provided in Figure 7).

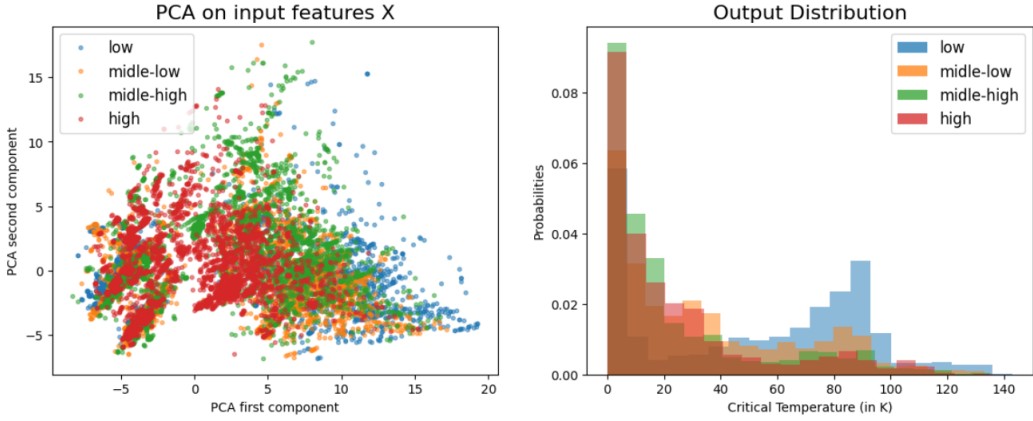

Figure 7: Visualization of the domain shift of the superconductivity data set. The visualization of the two first components of the PCA on the input features is given on the left. The output distribution is given on the right. One domain is represented by one color.

**Standard deviation** Table 2 presents the standard deviation of the MAEs obtained on the 8 repetitions of the 12 experiments with the Balanced Weighting training and $K = 20$. We observe that K-medoids provides the smallest standard deviations in the majority of the experiments. K-medoids is indeed a deterministic algorithm and thus selects a determined batch of target points. Besides, K-medoids selects the target batch to label in a distribution matching perspective, i.e. it produces a training set with a distribution close to the one of the testing set (cf Section 2). This can explain why the training is more stable with the training set provided by K-medoids.

Table 2: Standard deviations of the MAE on the unlabeled data for the superconductivity experiments. The deviations are computed using the results of 8 repetitions of each experiment.

| Experiment | l→ml | l→mh | l→h | ml→l | ml→mh | ml→h | mh→l | mh→ml | mh→h | h→l | h→ml | h→mh |
|---|---|---|---|---|---|---|---|---|---|---|---|---|
| Random | 1.007 | 2.599 | 1.597 | 1.232 | 1.679 | 1.291 | 1.863 | 0.787 | 0.266 | 1.900 | 1.228 | 0.595 |
| Kmeans | 1.05 | 0.93 | **0.78** | 1.93 | **0.54** | **0.64** | 1.0 | 1.36 | 0.58 | 6.51 | 1.51 | 0.59 |
| QBC | 1.469 | 2.370 | 1.882 | 0.571 | 0.564 | 0.707 | 2.151 | 0.767 | 0.394 | 7.240 | 3.250 | 0.620 |
| Kcenters | 0.807 | 1.761 | 1.926 | 0.626 | 0.343 | 0.794 | 2.188 | 0.885 | 0.503 | 5.403 | 2.838 | 0.920 |
| Diversity | 1.34 | 1.2 | 3.28 | **0.52** | 0.8 | 1.86 | 2.46 | 0.79 | 0.42 | 5.49 | 1.44 | 0.32 |
| Kmedoids | **0.59** | **0.76** | 1.21 | 0.54 | 0.55 | 0.74 | **0.64** | **0.33** | **0.26** | **1.19** | **0.8** | **0.15** |

Table 3: MAE on the critical temperature for the Superconductivity experiments with Balanced Weighting and $K = 20$.

| Experiment | l→ml | l→mh | l→h | ml→l | ml→mh | ml→h | mh→l | mh→ml | mh→h | h→l | h→ml | h→mh |
|---|---|---|---|---|---|---|---|---|---|---|---|---|
| Random | 15.33 | 15.80 | 17.45 | 16.53 | 11.39 | 14.70 | 17.65 | 12.83 | 10.36 | 18.75 | 14.86 | 10.54 |
| Kmeans | 14.43 | 13.60 | **13.98** | 15.79 | 10.19 | 12.73 | 17.18 | 12.67 | 10.02 | 22.10 | 14.69 | 9.76 |
| QBC | 20.00 | 19.03 | 20.08 | 15.89 | 12.24 | 15.31 | 20.78 | 12.87 | 10.19 | 31.88 | 18.86 | 10.65 |
| Kcenters | 19.21 | 15.73 | 16.85 | 15.75 | 11.62 | 13.44 | 22.17 | 12.74 | 10.24 | 36.50 | 19.60 | 10.39 |
| Diversity | 19.46 | 18.21 | 18.68 | 16.01 | 11.94 | 15.36 | 23.92 | 14.31 | 10.70 | 37.97 | 20.89 | 10.78 |
| Kmedoids | **12.70** | **13.57** | 14.11 | **14.49** | **10.02** | **12.52** | **15.36** | **12.37** | **9.79** | **16.62** | **14.14** | **9.32** |

**(high to low) analysis** An interesting fact to notice in the results of the superconductivity experiments (for which a recap is given in Table 3) is the asymmetry of results between the low-to-high experiment and the high-to-low experiment. Indeed, the results on the latter experiment are worse than the ones of the former, except for the Random and K-medoids algorithms.

It first appears, in Figure 8, that this difference in performance for the low-to-high experiment compared to the high-to-low experiment comes from the difference in the output distributions of the low and high domains. Indeed, the output distribution of the high domain is made of one mode concentrated in the low temperature while the output distribution of low is made of two modes, one in the low and the other in the high temperature. Thus, the model trained with the low domain data will generalize better to the high domain since the model has seen the full range of temperature. In the inverse problem (high-to-low), the model has only seen low temperatures and thus poorly predict a part of the low domain. Then, for this difficult experiment (high-to-low), the difference in performance between the different query algorithms lies mainly in the number of data with high temperature in the low domain that are queried (see Figure 9). It appears that K-medoids and Random query more of this type of data, as they are highly concentrated and relatively far from the sources (see Figure 10, 11).

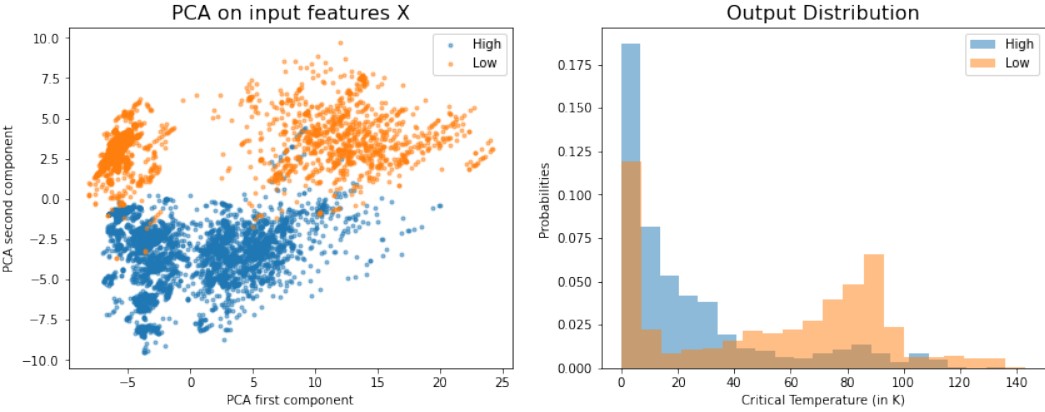

Figure 8: Visualization of the domain shift of the superconductivity data set for the high-to-low experiment. The visualization of the two first components of the PCA on the input features is given on the left. The output distribution is given on the right. One domain is represented by one color.

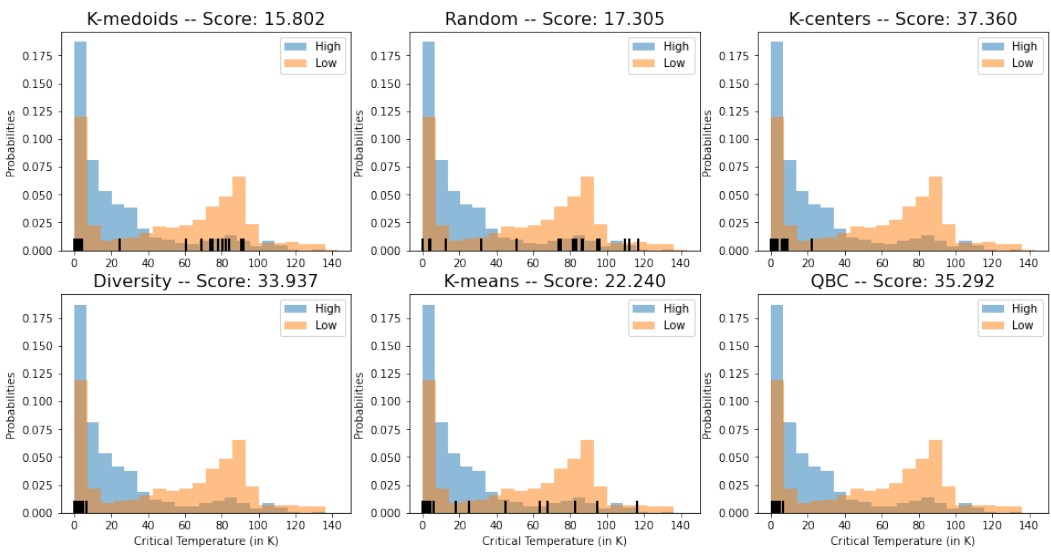

Figure 9: Visualization of the queries in the output space for the high-to-low experiment with $K = 20$.

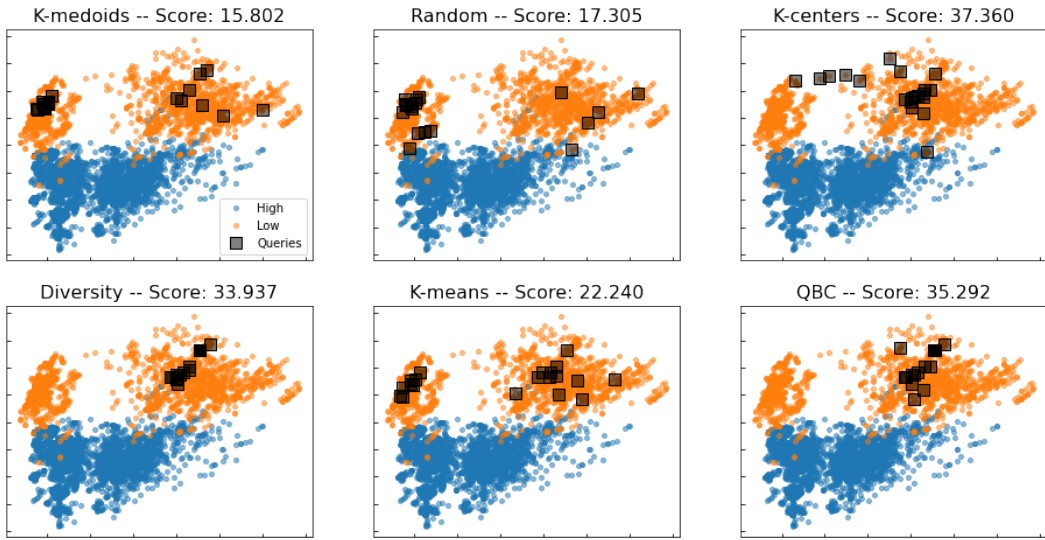

Figure 10: Visualization of the queries in the input space for the high-to-low experiment with $K = 20$.

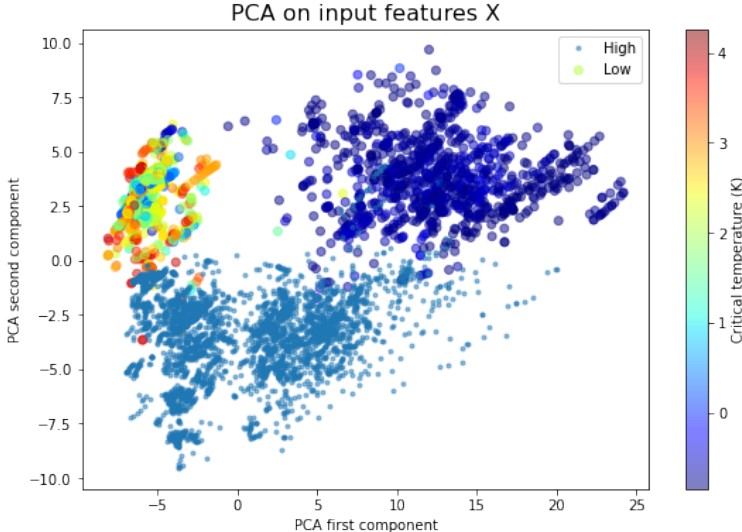

Figure 11: Visualization of the the input space for the high-to-low experiment. The heatmap gives the output labels for the "low" domain.

## G.2    OFFICE

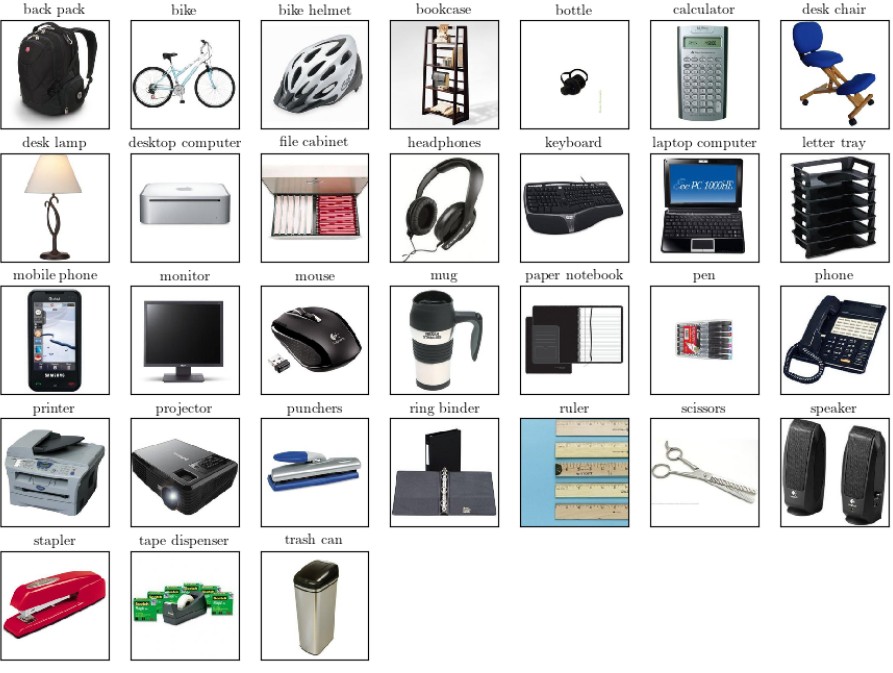

Figure 12: Office data set: examples of images from amazon domain

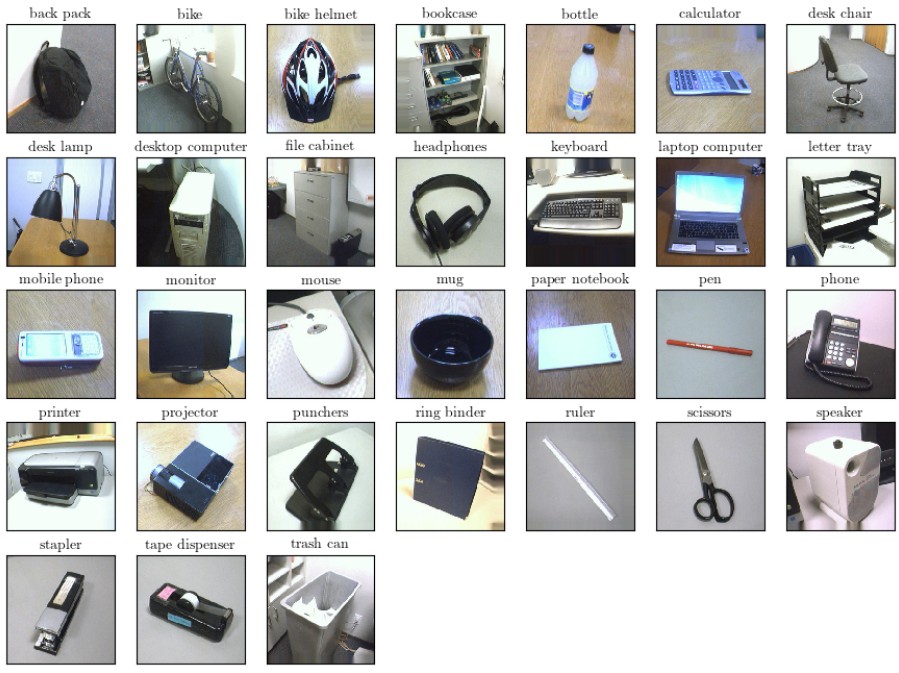

Figure 13: Office data set: examples of images from webcam domain

## G.3    DIGITS

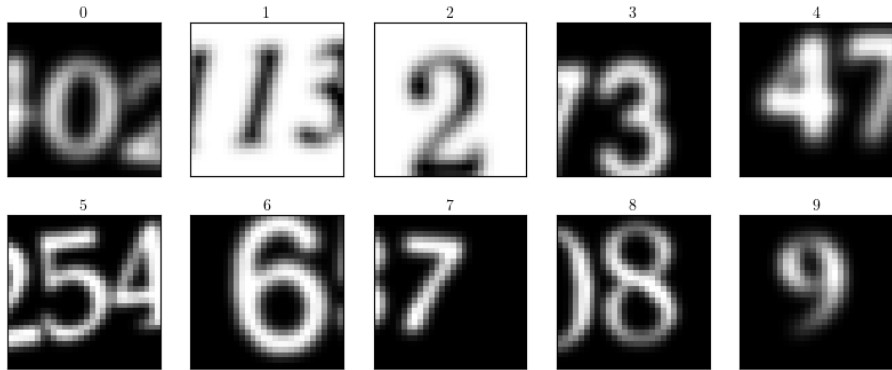

Figure 14: Digits data set: examples of SYNTH images

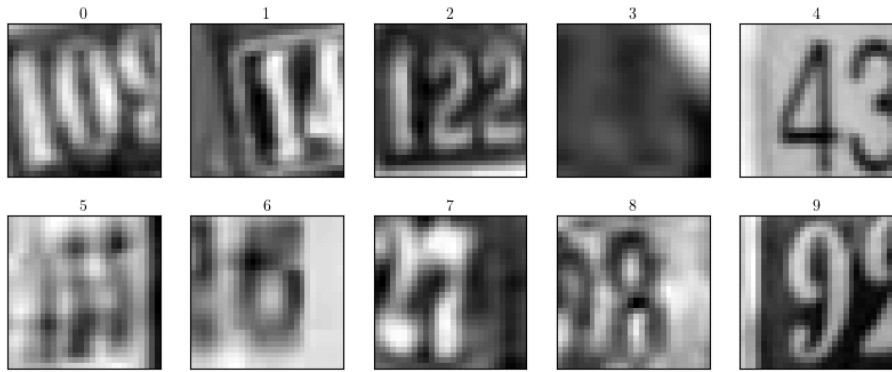

Figure 15: Digits data set: examples of SVHN images

