# OpenReview forum: "Discrepancy-Based Active Learning for Domain Adaptation"
_ICLR.cc/2022/Conference — ICLR 2022 Poster_

### Official Review · Reviewer_Xjwo · 2021-10-31

**Correctness:** 4
**Technical Novelty And Significance:** 2
**Empirical Novelty And Significance:** 3
**Recommendation:** 6
**Confidence:** 4

**Main Review:**

Strengths:
1. The authors provide a computable way to conduct batch active learning for domain adaptation, and the method shows empirical advantages over existing ones.
2. The authors provided an accelerated version of the K medoids algorithm, and also analyzed its computational complexity.
3. The paper is generally well-written, and the authors also provide visual insights to help understand the paper.

Weaknesses:
1. Since the localized discrepancy is proposed in Zhang et al 2020 (as noted by the authors in the paper as well), I think it's fair to say that, on the theoretical side, the main contributions are (1) the relaxation of the localized discrepancy under Lipschitzness and (2) the accelerated K-medoids algorithm (for the relaxation). That being said, the proposed algorithms only apply to restricted cases.
2. Following the previous point, my biggest concern is how well the relaxation is, even under Lipschitzness? The authors didn't provide a thorough comparison between bounds in Eq (3) and Eq (4). As a result, the relaxation could be meaningless if Eq (4) is much larger than Eq (3) (say, in some cases).
3. The algorithm is restricted to the batched active learning case, where the learner picks the queried data in a single batch. What would can in the more general sequential case?
4. I'm guessing that the requirement $\epsilon \geq \eta_H$ in the theoretical guarantees is to make $H^K_{\epsilon}$ is not empty, is that right? But I wonder why choose $\eta_H$ as the threshold value? Seems that we could have $H^K_{\epsilon}$ being non-empty for an $\epsilon$ smaller than $\eta_H$ since the definition of $\eta_H$ has a summation of two things.
5. In Section 4.3 the authors wrote "... the parameter $\epsilon$ will be small because ... This would lead to $\epsilon < \eta_H$ and the bound would not be valid anymore". Isn't it true that $\eta_H$ would get smaller as well when $H$ has a great approximation power. Also, a related question is that I thought $\epsilon$ is a parameter that the learner gets to choose, i.e., one can simply choose an $\epsilon$ larger than $\eta_H$ (if it's known).


**Summary Of The Paper:**

This paper studies active learning for domain adaption for a set of Lipschitz functions. The paper proposes to use a localized discrepancy to restrict the relevant candidate hypotheses. Under Lipschitzness, the localized discrepancy is further relaxed into a distance measure over the X domain; the authors also design an accelerated K-medoid algorithm to minimize such distance. In special cases (under certain simplifying assumptions), theoretical guarantees presented in this paper show advantages over previous ones. The proposed algorithm also shows empirical advantages over existing ones.

**Summary Of The Review:**

Based on the strengths and weaknesses listed above, I think overall the paper is marginally above the threshold.

---

> ### Author Response · Authors · 2021-11-24
> **Response to reviewer Xjwo**
>
> The authors are very grateful to the reviewer for making these very clear and detailed reviews which relevantly highlight both strength and weakness of the paper. Hereafter are the responses to the reviewer main concerns:
>
> 1, 2\. First, we thank the reviewer for noting this point about the relaxation between Eq (3) and Eq (4). In fact, the authors did make the theoretical study to compute the approximation error made by further bounding the localized discrepancy by the criterion of Eq (4). This study was not presented in the previous version for not overloading the paper, it has been added now in the new version in Appendix C.
>
> We show that the bounds of Eq (3) and Eq (4) are close at least in the case where the Lipschitz constant of the hypothesis space $k$ is sufficiently higher than the one of the labeling function $k_f$ (the approximation factor between the tow objectives being $k / (k-k_f)$).
>
> 3\. Any algorithm applied in the single batch case can be applied in the sequential case, as the former is more general. For what the author observe in practical scenario, especially in industrial cases, the labeling of new points are performed in a single batch as the labeling process is time consuming because of the use of heavy testing machines.
>
> 4\. Yes, indeed, the requirement $\epsilon \geq \eta_H$ is only a sufficient condition, the bound can still apply for $\epsilon < \eta_H$ but is not guarantee.
>
> 5\. It is true, indeed, that $\eta_H$ will also decrease, but if $f_Q$ and $f_P$ are different, epsilon can be greater than $\max_{x \in \mathcal{S} \cup \mathcal{T}} L(h(x), f_Q(x))$ but smaller than $\max_{x \in \mathcal{S} \cup \mathcal{T}} L(h(x), f_P(x))$. In this case $H_{\epsilon}^K$ does not contain the best target hypothesis anymore and the bound of Eq (3) should be written with $\eta_{H_{\epsilon}^K}$ instead of $\eta_H$.
>
> Notice that the parameter epsilon has been introduced for theoretical purpose, it reflects the error that the learner tolerates on the labeled data. For most  machine learning models, it is difficult to estimate but can be linked to different factors such as the complexity of the hypothesis set, the optimization algorithm used, the importance weighting apply on the labeled data. For instance, we can consider that using big neural networks fitted with a high number of epochs corresponds to small epsilon.

---

### Official Review · Reviewer_Nd1E · 2021-11-03

**Correctness:** 4
**Technical Novelty And Significance:** 3
**Empirical Novelty And Significance:** 3
**Recommendation:** 6
**Confidence:** 3

**Main Review:**

Studying the domain adaptive active learning problem on both theoretical and methodological levels in a consistent way is a strong side of the paper. On the other hand, the novelty of the methodological contribution is limited as the proposed adaptation of the K-medoids method is simply put forward as a combination of a couple of baseline strategies from previous literature. Another aspect that could be improved is the presentation of the proposed algorithm: The discussion of the proposed method is given at a very high level, without explaining, motivating or describing the involved steps in much detail.

More detailed remarks:

1. In Theorem 1:
- Please define explicitly what the distance d(x',x) stands for.
- "We denote by M the bound of L": Please define this mathematically.
- I guess the term L_P(h,f) should have been L_P(h,f_P) rather.

2. In the experimental results, there are some algorithms used in the comparisons in Figure 3, but skipped in the results of Figure 2 (like BADGE, CLUE, ...). Is there a reason for this (not suitable for regression tasks, etc.)?

**Summary Of The Paper:**

The paper studies the problem of active learning for domain adaptation. The paper has two main contributions. First, the problem of the selection of a query target sample set is studied theoretically. A generalization bound is presented, which is based on the recently proposed concept of localized discrepancy as opposed to rather classical discrepancy measures that can be too conservative. The second contribution of the paper is an algorithm for the batch selection of K target queries, which is motivated by the theoretical findings.

**Summary Of The Review:**

While the proposed methodology is of limited novelty and not described in a self-contained manner, when considered along with the proposed theoretical framework, the contributions of the paper may be worth sharing with the community.

---

> ### Author Response · Authors · 2021-11-24
> **Response to reviewer Nd1E**
>
> We thank the reviewer for the interesting comments and suggestions that help us to improve the writing of the paper.
>
> Essentially, the different steps of the K-medoids Accelerated algorithm are motivated by the kind of problem we encounter when making active learning for domain adaptation. First, the existence of a large set of preselected medoids (corresponding to the source points) require an efficient initialization of the minimal distances between target points and the labeled set. As only the minimal distance is required, we find that KDT-Forest-Nearest-Neighbour is suited for this problem. However, this same algorithm can not be used for the computation of pairwise distances between target points. This fact motivated the use of an iterative algorithm of nearest medoid assignment (forming cluster) and medoid update in each cluster. But, as a cluster can still contain many target points, we propose a branch and bound algorithm to reduce the number of computation. This algorithm is fully described in Algorithm 3 in Appendix E.1. It relies on a statistical threshold: the individual criterion for each target point in the cluster are estimated with a small batch of target points. Then, if one estimated criterion is out of the confidence interval of the minimal estimated criterion, we stop to estimate this criterion. As far as the author know, this branch and bound approach for medoid update is an interesting novelty.
>
> Concerning the more detailed remarks:
>
> 1. The authors thank the reviewer for these comments which have been taken into account in the new paper version.
>
> 2. Indeed, CLUE and BATCH as well as AADA and BVSB are active learning algorithms designed for classification tasks as they rely on the probabilities of belonging to the different classes given by a predicter.

---

> > ### Comment · Reviewer_Nd1E · 2021-11-29
> > **Thanks**
> >
> > I would like to thank the authors for their response to my comments.

---

### Official Review · Reviewer_bRBm · 2021-11-04

**Correctness:** 3
**Technical Novelty And Significance:** 2
**Empirical Novelty And Significance:** 2
**Recommendation:** 6
**Confidence:** 4

**Main Review:**

Pros:

1) Overall, the paper addresses an important problem and formalizes the task of active learning for domain adaptation.
2) It is interesting to see that the minimization of the target risk maps to the k-medoid problem.

Cons:

1) One of the main concerns is that the final algorithm used is a classical clustering algorithm based on k-medoids. K-medoids is well studied and has been used in the active learning context before and several efficient implementations of the k-center have been studied.

2) In addition to datasets such as UCI and superconductivity, it would be good to show results on standard datasets such as CIFAR10, CIFAR100, and MNIST by treating one as source and the other one as target?

3) While the paper says that k-medoid can be solved using greedy and provides a bound of 1-1/e it does not explain how this is achievable. One possible interpretation is that k-medoid can be seen as the maximization of submodular functions and this leads to an approximation guarantee of 1-1/e. The bound (1-1/e)-approximation only applies to the original objective function of k-medoid where we use greedy to compute all the elements of the subset. Here we use all the points from the source set and only label K elements in the target set. The authors need to carefully provide all the details of this bound as it may not directly translate from the original problem.

4) There is also a close connection to the use of MMD for subset selection shown in [Kim et al. 2016]. This paper looks at the MMD measure and shows that the selection of prototypes and criticisms can be mapped to the maximization of submodular functions and solved using greedy as well. Kim et al. Examples are not enough, learn to criticize! Criticism for Interpretability, 2016.

5) I find this comparison with k-center a bit vague. The paper makes simplifying assumptions such as f=f_P = f_Q and \epsilon = 0 and shows that the bound using k-medoids is tighter than k-center in equations (5) and (6).  While max_{x’\in T} d(x’,L_k) is higher than (1/n)\sum_{x’ \in T}d(x’L_k), both k-center and k-medoids are solved with greedy algorithms with approximation guarantees, and it is not clear whether such a comparison can be treated as a formal result.

6) It would be good to provide more intuition on the choice of weighted k-medoids algorithm with weights multiplied with the min d(x,x’)? Typically, when we do a weighted extension, we can consider two terms in many active learning objective functions where one term models the uncertainty that looks for points near the decision boundary and the other term models the uncertainty. The paper uses uncertainty weights loosely in equation (7) without sufficient justification as to why multiplying the weights would be better than using this as an additive term.


**Summary Of The Paper:**

This paper proposes a k-medoid solution for active learning in the context of domain adaptation. The paper builds on top of Mansour et al. (2009) and looks at the discrepancy between source and target distributions. Looking at the whole hypothesis space is very conservative since this would include hypotheses that the learner would never consider as a labeling function. In order to deal with this problem, this paper only considers localized discrepancy (Zhang et al. 2020) where we only consider the hypotheses that are epsilon away from labeling function for source domain, i.e. we are only considering labeling functions that are epsilon away from the source labeling function. Generalization bounds are derived using Rademacher average and localized discrepancy for general loss functions. From these bounds, the paper shows that one can minimize the target risk by solving a k-medoid problem.


**Summary Of The Review:**

While the formalizes and motivates the proposed solution, I don't see any novelty in the final algorithm used for domain adaptation. There are also many loose ends that I feel that the authors should address.

---

> ### Author Response · Authors · 2021-11-24
> **Response to reviewer bRBm**
>
> The authors thank the reviewer for these detailed and thorough reviews. The following paragraph response to the reviewer main concerns:
>
> 1\. K-medoids has, indeed, been used for active learning in several works which are referenced in the related work (Section 4.1). The authors didn't claim the novelty of using this algorithm but derive novel theoretical guarantees on the target risk for K-medoids by highlighting its link with the localized discrepancy. Besides, although several algorithms exist to solve or approximate the K-medoids (cf Section 3), the authors didn't find an algorithm which respond to their specific needs. Indeed, when dealing with relatively large data set, the algorithm starts with a large set of medoids (the source points) which require to carefully deal with the initialization and the medoid assignment. Notice finally that the developed branch and bound algorithm for the medoid update in each cluster, based on statistical threshold, is, as far as we know, an interesting novelty.
>
> 2\. Please notice that two experiments have been conducted on standard data sets for domain adaptation: Synth vs SVHN [Ganin et al., 2016] (which are digits data sets like MNIST) and Office [Saenko et al., 2010] (which is a data set of real images like CIFAR).
>
> 3\. We are very thankful to the reviewer for this relevant remark. Indeed, the authors were thinking about the results on submodular functions. Notice that this results is applied on the gain of the K-medoids algorithm, i.e. the amount of decrease of the objective by adding $K$ medoids:
>
> $\text{gain}(\mathcal{T_{\textit{K}}}) = \sum_{x' \in \mathcal{T}} d(x', \mathcal{S}) - \sum_{x' \in \mathcal{T}} d(x', \mathcal{S} \cup \mathcal{T_{\textit{K}}})$
>
> This was unclear in our statement and has been clarified in the new paper version. As the reviewer suggested, we also add the details of the bound for our case where a subset in greedily added to the sources (see Appendix E.2).
>
> 4\. Making the link between our work and [Kim et al., 2016] is indeed very interesting and the reference has been added to the related work. Kim et al. use the MMD as a measure for distribution matching, they consider reproducing kernel Hilbert space as hypothesis space and derive an objective function for subset selection which is shown to be monotone submodular. Their approach is different from ours as they consider a different measure for distribution matching, hypothesis set and objective function. We notice, however, that they evoke in their conclusion the plan of extending the algorithm to large data set. Our proposed accelerated method may also applied to their algorithm and have then a larger application than for K-medoids.
>
> 5\. The simplifying assumptions $f=f_P=f_Q$ and $\epsilon = 0$ are the ones that are made in [Sener and Savarese, 2018] for their bounds. We compare our proposed bound using their settings.
>
> It is true that because of the approximation algorithms used by both methods, it may be some cases where using greedy for K-centers leads to smaller criterion than K-medoids. A sentence to mitigate the theoretical comparison has been added in the new version. We propose an empirical comparison of the objectives of K-medoids and K-centers for the Digits experiments (see Appendix F, Figure 6). We observe in this case that the K-medoids objective is smaller than the one of K-centers.
>
> 6\. The intuition is to make a preselection of target points by selecting the ones that are in the margin between classes as they are the most interesting points to label [Balcan et al., 2007]. A target point $x' \in \mathcal{T}$ which is not in the margin may be far from the source points in term of the distance $d$ in the input space, but as the common loss functions for classification are bounded (by $1$ for instance), bounding the maximal potential error in $x'$ by $2 k \mu \\, d(x', \mathcal{S})$ is overestimated when $1 << 2 k \mu \\, d(x', \mathcal{S})$. Besides, as the prediction for $x'$ is given with high confidence, it is very unlikely that it could be misclassified.
>
> This consideration explains why we choose to multiply the weights to the distances instead of summing the spatial criterion to the confidence criterion. Indeed, a target point far from the source ($d(x', \mathcal{S}) >> 1$) but predicted with high confidence $\text{bvsb}(x')=0$, will not impact the objective criterion with the multiplying option as $\text{bvsb}(x')*d(x', \mathcal{S})=0$ (which is expected) but will still highly drives the medoid selection with the summing option $\text{bvsb}(x')+d(x', \mathcal{S}) >> 1$.
>
> References:
>
> Ganin et al., Domain-adversarial training of neural networks, 2016
>
> Saenko et al., Adapting visual category models to new domains, 2010
>
> Kim et al., Examples are not enough, learn to criticize! criticism for interpretability, 2016
>
> Sener and Savarese, Active learning for convolutional neural networks: A core-set approach, 2018
>
> Balcan et al, Margin based active learning, 2007

---

> > ### Comment · Reviewer_bRBm · 2021-11-29
> > **Response to authors**
> >
> > We thank the authors for carefully addressing the concerns regarding bounds, and comparison with k-center. I have increased my score.

---

### Official Review · Reviewer_jrGL · 2021-11-08

**Correctness:** 3
**Technical Novelty And Significance:** 3
**Empirical Novelty And Significance:** 2
**Recommendation:** 6
**Confidence:** 3

**Main Review:**

### strength

- In the conventional active learning methods, it has been the mainstream to select the input of the target domain to be labeled based on the output of the domain classifier trained at the same time (in general, the relationship between this criterion and the expected risk of the target domain is unclear). On the other hand, this paper proposes an single-shot active learning method that chooses the data to be labeled in the target domain in such a way that the expected risk of the target domain is directly reduced.

- The part corresponding to the discrepancy in the theoretically derived upper bound of the target risk does not depend on the complexity of the hypothesis set (strength of Lipschitz condition) and the shape of the loss function. This has the advantage that the algorithm for active learning can be constructed in the way of hypothesis set-free or loss function-free. Actually, the proposed algorithm can be interpreted as a clustering algorithm that minimizes the sum of the distances between the data.

- The proposed method, Accelerated K-medoids, is not only a method for selecting K inputs to be labeled in the target domain, but also an algorithm for clustering. In fact, it is an improved version of Greedy K-medoids method in which the computational complexity with respect to sample size is improved from n^2 to n^3/2.


### weakness
- In the appendix, the authors compare the empirical computational time of the Greedy K-medoids and the Accelerated one, and show the effectiveness of the latter.　On the other hand, it is not specifically mentioned how these two algorithms relate in terms of the expected risk of the target domain. If the two algorithms produce identical output, it is better to explain that, and if not, the two algorithms should be compared not only in terms of computational time but also in terms of the expected risk of the target domain.


### questions
- In the results shown in Table 1, the accuracy of l → h (low to high) and h → l (high to low) seem to change significantly (the latter is worse) except for random and Kmedoids. What could be the reason for this? Also, can you argue that the proposed method gives more robust results even when this phenomenon (asymmetry) occurs using other methods?

- For the Localized Discrepancy, how much can we expect the situation that "the restricted hypothesis set H^K_{\epsilon} is not empty and the tolerance \epsilon is reasonably small"?

- Although this paper deals with 1-shot active learning, can I understand that if we iterate this method, we can also empirically approximate the risk in the target domain and evaluate the joint error \eta_H in the risk upper bound?

- In Sec 5.1, the author claim that "We assume that the architecture and the resulting hyper-parameters will still be appropriate after adding the queried target data to the training set" while could you give an example of possible situations where this assumption does not hold? I would like to know how reasonable this assumption is.

**Summary Of The Paper:**

This paper proposes an active learning method to efficiently select data in the target domain that is suitable for learning hypotheses together with data in the source domain in domain adaptation problems.

The authors first propose to evaluate the dissimilarity between the source and target domains by localized discrepancy, where the hypothesis set is restricted to includes only hypothesithes with sufficiently small errors with the label function of the source domain on the training data (the union set of the input of the source domain and the input of the target domain to be labeled). Then an upper bound of the expected risk for the target domain based on this discrepancy. The main theoretical contribution is that the authors have shown that the discrepancy term in this upper bound is further bounded from above by the sum of the distances between the inputs of the training data and the inputs of the target domain, and this is the background for the design of the active learning algorithm.

In domain adaptation, the goal is to solve the problem of minimizing the discrepancy between the two domains. From the statements of this theorem, it can be replaced by the problem of selecting K data in the target domain so as to minimize the sum of the distances between the inputs of the training data and the inputs of the target domain. The latter problem is equivalent to solving the K-medoids problem for clustering, and the authors' technical contribution is that they have proposed an accelerated algorithm for this problem.

Finally, the authors conduct an evaluation experiment of the proposed method under various scenarios of domain shift using three types of benchmark data.


**Summary Of The Review:**

This paper deals with the important topic of active transfer learning and is considered to be a contribution to both theory and experiment.　Although there are some shortcomings in the experiments and some questions, I can generally support the acceptance of the paper.

---

> ### Author Response · Authors · 2021-11-24
> **Response to reviewer jrGL**
>
> The authors are very thankful to the reviewer for the relevant remarks and suggestions which really help us to improve the paper.
>
> First, concerning the remark on the comparison of empirical computation time, the reviewer made the relevant observation that the two algorithms K-medoids Greedy and K-medoids Accelerated can select different sets of points. The question is therefore: does the saving of computational time achieved by K-medoids Accelerated comes at the expense of a worse subset selection? As suggested by the reviewer, the author computed the objective of each algorithm for different numbers of queries and showed the results in Figure 6 in Appendix F. We first observe that the K-centers objective is much higher than the one of K-medoids Greedy and K-medoids Accelerated, which comforts the theoretical considerations of Section 4.2. We also observe that the K-medoids Greedy objective is slightly lower than the one of K-medoids Accelerated when the number of queries increases. Thus, the learner may prefer to use the Greedy algorithm rather than the accelerated version if computational time is not a limitation, otherwise, the significant gain in computational time of the accelerated algorithm is worth the price of approximation.
>
> Note that a theoretical comparison of the two algorithms in terms of performance is not obvious because it involves comparing a minimum on the entire target data set to a minimum on a batch of size $B$. For example, for $K=1$, if one want to compare the objective of K-medoids Greedy versus the one of K-medoids Accelerated, one needs to compare the following two quantities:
>
> $\min_{x_0' \in \\mathcal{T}} \sum_{x' \in \mathcal{T}} \min_{x \in \\mathcal{S} \cup \\{x_0' \\}} d(x', x)$
>
> $\min_{x_0' \in \mathcal{B}} \sum_{x' \in \mathcal{B}} \min_{x \in \mathcal{S} \cup \\{x_0' \\}} d(x', x)$
>
> With $\mathcal{B} \subset \mathcal{T}$ a batch of size $B < n$ picked randomly uniformly from $\mathcal{T}$.
>
> Without any assumptions on the distribution of the target points, the best medoid $x_0^* \in \mathcal{T}$ may have a very low objective compared to the second best medoid. Then, if $x_0^*$ is not chosen in the batch $\mathcal{B}$, the difference in objective can be very large.
>
> Concerning the different questions:
>
> 1. The authors thank the reviewer for noticing this fact, which is indeed very interesting. We have done additional research to understand what is happening in this case. The observations are described in detail in Appendix G.1. Here is a brief summary of what has been observed:
> First, it appears that the difference in performance for the low-to-high experiment compared to the high-to-low experiment comes from the difference in the output distributions of the low and high domains (see Figure 8). Indeed, the output distribution of the high domain is made of one mode concentrated in the low temperatures while the output distribution of low is made of two modes, one in the low and the other in the high temperatures. Thus, the model trained with the low domain data will generalize better to the high domain since the model has seen the full range of temperatures. In the inverse problem (high-to-low), the model has only seen low temperatures and thus poorly predict a part of the low domain.
> Then, for this difficult experiment (high-to-low), the difference in performance between the different query algorithms lies mainly in the number of data with high temperature in the low domain that are queried (see Figure 9). It appears that K-medoids and Random query more of this type of data, as they are highly concentrated and relatively far from the sources (see Figures 10, 11).
>
> 2. This is an interesting question. It depends on what is considered to be "reasonably small" for epsilon. The authors think that, in the case where the learner is able to produce a hypothesis close to the labeling function on the labeled data, the epsilon can be small and $H_{\epsilon}^K$ be non empty.
>
> 3. The $\eta_H$ parameter is related to the $\lambda$ parameter in [Zhang et al., 2019] and the $\eta_H$ parameter in [Cortes et al., 2019]. As far as we understand, these parameters are used for theoretical purpose and are not estimated in the algorithms. That said, it could be of some interest to investigate in future works, how this parameter could be estimated in a sequential selection process and how its estimation could help the next selections.
>
> 4. We can consider, for example, the problem where the source input distribution $Q$ is a Gaussian distribution $\mathcal{N}(-3, 1)$ and the target input distribution is a Gaussian distribution $\mathcal{N}(0, 1)$ with $f_Q, f_P: x \to |x|$. In this case, the learner can consider using a shallow neural network with one neuron and a Relu activation function, which is sufficient to learn the task on the source domain. However to learn the task on both domains, it would require to use a neural network with at least two neurons in the hidden layer.

---

### Decision · Program_Chairs · 2022-01-20

**Decision:**

Accept (Poster)

**Comment:**

This paper deals with the important topic of active transfer learning. All reviewers agree that
while the paper presents some shortcomings , it is considered to be a worth contribution.